# MixSup: Mixed-grained Supervision for Label-efficient LiDAR-based 3D Object Detection

**Yuxue Yang**[1,2,3,5]    **Lue Fan**[2,3,5†]    **Zhaoxiang Zhang**[1,2,3,4,5†]

[1]School of Artificial Intelligence, UCAS

[2]University of Chinese Academy of Sciences (UCAS)

[3]Institute of Automation, Chinese Academy of Sciences (CASIA)

[4]Centre for Artificial Intelligence and Robotics (HKISI_CAS)

[5]State Key Laboratory of Multimodal Artificial Intelligence Systems (MAIS)

{yangyuxue2023,fanlue2019,zhaoxiang.zhang}@ia.ac.cn

## Abstract

Label-efficient LiDAR-based 3D object detection is currently dominated by weakly/semi-supervised methods. Instead of exclusively following one of them, we propose **MixSup**, a more practical paradigm simultaneously utilizing massive cheap coarse labels and a limited number of accurate labels for **Mix**ed-grained **Sup**ervision. We start by observing that point clouds are usually *textureless*, making it hard to learn semantics. However, point clouds are *geometrically rich* and *scale-invariant* to the distances from sensors, making it relatively easy to learn the geometry of objects, such as poses and shapes. Thus, MixSup leverages massive coarse *cluster-level* labels to learn semantics and a few expensive *box-level* labels to learn accurate poses and shapes. We redesign the label assignment in mainstream detectors, which allows them seamlessly integrated into MixSup, enabling practicality and universality. We validate its effectiveness in nuScenes, Waymo Open Dataset, and KITTI, employing various detectors. MixSup achieves up to **97.31**% of fully supervised performance, using cheap cluster annotations and only 10% box annotations. Furthermore, we propose **PointSAM** based on the Segment Anything Model for automated coarse labeling, further reducing the annotation burden. The code is available at https://github.com/BraveGroup/PointSAM-for-MixSup.

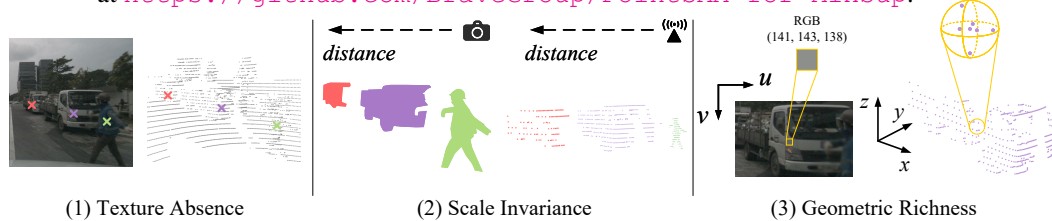

| (1) Texture Absence | (2) Scale Invariance | (3) Geometric Richness |
|---|---|---|

Figure 1: **Illustration of distinct properties of point clouds compared to images.** They make semantic learning from points difficult but ease the estimation of geometry, which is the initial motivation of MixSup.

## 1 Introduction

LiDAR-based 3D perception is an indispensable functionality for autonomous driving. However, the laborious labeling procedure impedes its development in academia and industry. Therefore, many label-efficient learning approaches have emerged for LiDAR-based 3D object detection, such as semi-supervised learning (Zhao et al., 2020; Wang et al., 2021; Yin et al., 2022a; Liu et al., 2023a) and weakly supervised learning (Qin et al., 2020; Meng et al., 2020; 2021; Zhang et al., 2023b; Xia et al., 2023).

In this paper, we propose a more practical label-efficient learning paradigm for LiDAR-based 3D object detection. Particularly, we leverage massive cheap coarse labels and a limited number of

---

[†]Corresponding authors

accurate labels for **mix**ed-grained **sup**ervision (**MixSup**), instead of exclusively following one of the previous label-efficient learning paradigms. MixSup stems from our following observations of point clouds. (1) *Texture absence*: 3D point cloud lacks distinctive textures and appearances. (2) *Scale invariance*: point clouds in the 3D physical world are scale-invariant to the distance from sensors since there is no perspective projection like 2D imaging. (3) *Geometric richness*: consisting of raw Euclidean coordinates, the 3D point cloud naturally contains rich geometric information. We summarize these distinct properties in Fig. 1. These properties cut both ways. On the one hand, the lack of textures and appearances makes it challenging to learn the categories of point clouds and identify the approximate regions where objects are located, which are collectively referred to as *semantics*. On the other hand, scale invariance and geometric richness potentially make it relatively easy to estimate *geometric* attributes of objects, such as accurate poses and shapes.

Therefore, we derive the motivation of MixSup: *A good detector needs massive semantic labels for difficult semantic learning but only a few accurate labels for geometry estimation*. Fortunately, object semantic labels can be coarse and are much cheaper than geometric labels since the former do not necessitate accurate poses and shapes. So, in particular, we opt for semantic point clusters as coarse labels and propose MixSup aiming to simultaneously utilize cheap *cluster-level* labels and accurate *box-level* labels. Technically, we redesign the center-based and box-based assignment in popular detectors to ensure compatibility with cluster-level labels. In this way, almost any detector can be integrated into MixSup. To further reduce annotation cost, we utilize the emerging Segment Anything Model (Kirillov et al., 2023) and propose **PointSAM** for coarse cluster label generation, enjoying the "freebie" from the advances of image recognition. Our contributions are listed as follows.

1. Based on the observations of point cloud properties, we propose and verify a finding that a good detector needs massive coarse semantic labels for difficult semantic learning but only a few accurate geometric labels for geometry estimation.

2. We propose to adopt semantic point clusters as coarse labels and build a practical and general paradigm MixSup to utilize massive cheap cluster labels and a few accurate box labels for label-efficient LiDAR-based 3D object detection.

3. We leverage the Segment Anything Model and develop PointSAM for instance segmentation, achieving automated coarse labeling to further reduce the cost of cluster labels.

4. Extensive experiments on three benchmarks and various detectors demonstrate MixSup achieves up to 97.31% performance of the fully-supervised counterpart with 10% box annotations and cheap cluster annotations.

## 2 RELATED WORK

**LiDAR-based 3D Object Detection**   The mainstream LiDAR-based 3D detection can be roughly categorized into point-based methods and voxel-based methods. Point-based detectors (Shi et al., 2019; Yang et al., 2020; Shi et al., 2020b; Li et al., 2021) generally employ PointNet series (Qi et al., 2017a;b) as the point feature extractor, following diverse architectures to predict 3D bounding boxes. Voxel-based approaches (Zhou & Tuzel, 2018; Yan et al., 2018; Yin et al., 2021; Fan et al., 2022a;b; Chen et al., 2023b; Wang et al., 2023a;b; Liu et al., 2023d) transform raw points into 3D voxels, which facilitates 3D sparse convolution or transformer regimes. Besides, hybrid methods (Yang et al., 2020; Shi et al., 2020a; 2023) are utilized to harness the benefits from both sides.

**Semi-supervised Learning in 3D**   Semi-supervised learning aims to reduce the annotation burden by training models with a small amount of labeled data and a large amount of unlabeled data. Inspired by the achievement in 2D, semi-supervised learning has been propagated into 3D domain. SESS (Zhao et al., 2020) inherits the Mean Teacher (Tarvainen & Valpola, 2017) paradigm and encourages consensus between the teacher model and the student model. 3DIoUMatch (Wang et al., 2021) focuses on improving the quality of pseudo labels with a series of handcrafted designs. Different from 3DIoUMatch, Proficient Teacher (Yin et al., 2022a) leverages the spatial-temporal ensemble module and clustering-based box voting module to enhance the teacher model and obtain the accurate pseudo labels, removing the deliberately selected thresholds. Considering the weak augmentation in the teacher-student framework, HSSDA (Liu et al., 2023a) proposes shuffle data augmentation to strengthen the training of the student model.

**Weakly Supervised Learning** Weakly supervised learning employs inexpensive weak labels to mitigate the burden of annotation costs. Especially for outdoor scenes, the emerged methodologies mainly leverage weak annotations including click-level (Meng et al., 2020; 2021; Liu et al., 2022; 2023b; Zhang et al., 2023b), scribble-level (Unal et al., 2022) and image-level (Qin et al., 2020). Albeit these works achieve promising performance, they inevitably involve intricate training regimes or elaborate network architecture. In this paper, we find utilizing a few accurate labels can estimate good geometry. So it might be more practical to introduce some accurate labels instead of following a purely weakly-supervised setting.

## 3 PILOT STUDY: WHAT REALLY MATTERS FOR LABEL EFFICIENCY

In Sec. 1, we argue that a good detector needs massive coarse labels for semantic learning but only a few accurate labels for geometry estimation. Here we conduct a pilot study to confirm the validity of our claim.

We utilize predictions from a pre-trained detector (Fan et al., 2022b) to crop point cloud regions. Thus these regions are well-classified and we only need to focus on the objects' geometry estimation in the cropped regions. Before cropping, we introduce strong noise to the proposals to avoid geometry information leakage. In particular, we expand the proposals by 2 meters in all three dimensions, randomly shift them $0.2 \sim 0.5$ meters, and rotate them $-45° \sim 45°$. In this way, we build a well-

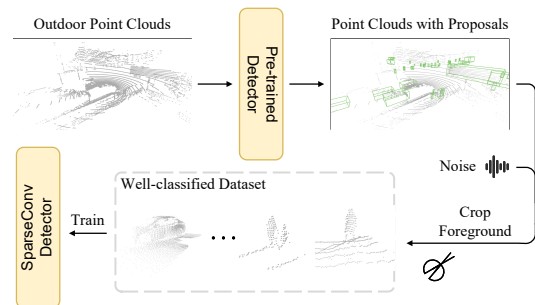

Figure 2: **Illustration of the pilot study.** We develop a well-classified dataset to factor out the classification and only focus on the influence of varying data amounts on geometry estimation.

classified dataset that comprises the cropped noisy regions. Finally, we train a sparse convolution based detector with different portions of the well-classified dataset. The illustration of the pilot study is shown in Fig. 2.

Table 1: Performances with varying data amounts on the well-classified dataset.

| Data amount | *Vehicle* 3D L2 AP / APH | | *Pedestrian* 3D L2 AP / APH | | *Cyclist* 3D L2 AP / APH | |
|---|---|---|---|---|---|---|
| | IoU = 0.7 | IoU = 0.5 | IoU = 0.5 | IoU = 0.3 | IoU = 0.5 | IoU = 0.3 |
| 100% | 64.19 / 63.74 | 81.70 / 80.86 | 65.23 / 58.02 | 76.77 / 67.65 | 67.04 / 65.99 | 70.94 / 69.71 |
| 20% | 64.02 / 63.54 | 81.60 / 80.74 | 65.00 / 58.04 | 76.56 / 67.68 | 66.89 / 65.85 | 71.09 / 69.88 |
| 10% | 63.37 / 62.89 | 81.50 / 80.60 | 64.78 / 57.96 | 76.56 / 67.81 | 66.26 / 65.14 | 70.55 / 69.24 |
| 5% | 63.38 / 62.90 | 81.45 / 80.54 | 64.11 / 56.73 | 76.16 / 66.76 | 65.38 / 64.21 | 70.49 / 69.12 |
| 1% | 56.40 / 55.75 | 79.01 / 77.51 | 57.92 / 50.26 | 73.06 / 62.63 | 55.85 / 54.63 | 60.18 / 58.70 |

The results in Table 1 show that performance with data amounts from 5% to 100% are quite similar. This phenomenon suggests that *LiDAR-based detectors indeed only need a very limited number of accurate labels for geometry estimation*. Additionally, we explore the impact of varying data amounts on the 3D detector's semantic learning in Appendix A.2, supporting our claim that *massive data is only necessary for semantic learning*. Fortunately, semantic annotations are relatively cheap and do not necessitate accurate geometry. So in the rest of this paper, we delve into the utilization of massive cheap coarse labels for semantic learning and limited accurate labels for geometric estimation.

## 4 METHOD

In this section, we first propose utilizing cluster-level labels and compare them with prior coarse center-level labels (Sec. 4.1) and how to integrate the coarse labels into MixSup for general use (Sec. 4.2). Then, we elaborate on how to obtain the coarse labels with PointSAM to further release the annotation burden (Sec. 4.3).

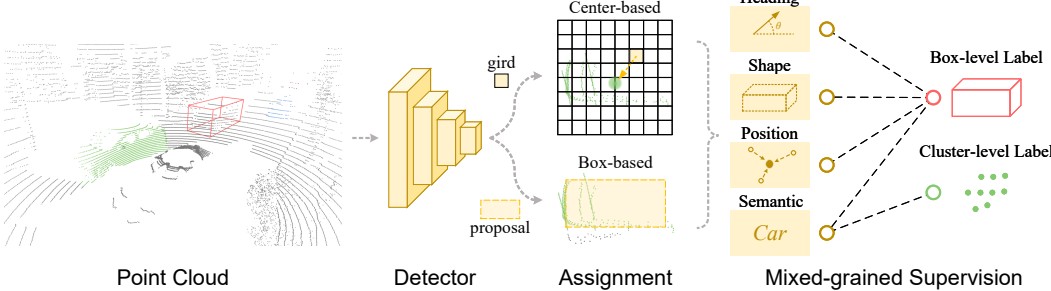

Figure 3: **Overview of MixSup.** The massive cluster-level labels serve for semantic learning and a few box labels are used to learn geometry attributes. We redesign the label assignment to integrate various detectors into MixSup.

## 4.1 CLUSTER-LEVEL COARSE LABEL

Obtaining precise 3D bounding boxes is a demanding and time-consuming undertaking, necessitating meticulous fine-tuning to meet the need for high-level accuracy. A line of work has emerged to acquire cheaper coarse labels, such as center-level labels (Meng et al., 2020; 2021). They click the center of objects on the Bird's Eye View to obtain center-level labels. Although straightforward, a single center point provides very limited information about an object and makes it inconvenient to adopt various types of detectors. In addition, it is also non-trivial for annotators to make an accurate center click.

Henceforth, we introduce clusters serving as better coarse labels. The acquisition of cluster labels is quite simple. Basically, annotators could follow this protocol: Making three coarse clicks around an object in Birds Eye's View. Then the three click points form a parallelogram serving as three corners of the parallelogram. The points inside the parallelogram form a coarse cluster. We emphasize the labeling of clusters is very efficient since it only needs three coarse clicks around the object corners instead of an accurate click in an exact object center. In Sec. 5.3, we empirically find the average labeling cost of a cluster is only around 14% of an accurate box. We provide a simple illustration of the labeling protocol in Appendix D.1.

## 4.2 COARSE LABEL ASSIGNMENT

In this subsection, we demonstrate how to integrate coarse cluster-level labels and box labels into different types of detectors for mixed-grained supervision, as illustrated in Fig. 3. The most relevant part to the labels in a detector is the label assignment module, responsible for properly assigning labels to the detector to provide classification and regression supervision. Thus, MixSup only needs to redesign the label assignments for cluster-level labels to ensure the generality. We categorize these assignments into two types: **center-based** assignment and **box-based** assignment.

**Center-based Assignment and Inconsistency Removal**   The center-based assignment is widely adopted in numerous detectors. For them, we substitute the original object centers with the cluster centers $\bar{c}$, which is defined in Eq. 1. The substitution inevitably leads to the inconsistency between the true object center (of accurate boxes) and the cluster center. To resolve the inconsistency, for box labels, we also use its inside cluster center as the classification supervision. As for regression supervision, it is only attained from a few box labels.

$$\bar{c} = \{\frac{\min \mathbf{x} + \max \mathbf{x}}{2}, \frac{\min \mathbf{y} + \max \mathbf{y}}{2}, \frac{\min \mathbf{z} + \max \mathbf{z}}{2}\}, \tag{1}$$

where $\mathbf{x}, \mathbf{y}, \mathbf{z}$ indicate the coordinate set of points in a cluster.

**Box-based Assignment**   Box-based assignment is the procedure of assigning labels to pre-defined anchors or proposals. For example, anchor-based methods consider anchors that have a high intersection over union (IoU) with box labels as positive. Similarly, two-stage methods select proposals having proper IoU with box labels for refinement and confidence learning. Below we only focus on assigning cluster-level labels to proposals, as the design for anchors is the same.

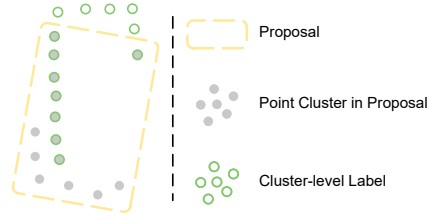

Figure 4: **Illustration of Box-cluster IoU.**

In order to implement box-based assignment, we first define box-cluster IoU, which is defined as the point-level IoU between point clusters in proposals and cluster-level labels. As depicted in Fig. 4, the gray dots represent the point cluster in the box, while the dots outlined in green denote the cluster-level label. The box-cluster IoU is computed as the ratio of the gray dots with green outlines to all the dots in the figure. With box-cluster IoU, we can assign cluster-level labels to proposals to train any anchor-based detectors and two-stage detectors.

**Ambiguity of Box-based Assignment**  It is worthwhile to note that the box-cluster IoU is essentially ambiguous. In particular, slight perturbations of bounding boxes can result in significant changes in the ordinary box IoU. However, slight perturbations on bounding boxes usually do not change the internal cluster, so box-cluster IoU may remain unchanged. Fortunately, we only rely on box-cluster IoU for semantic assignment instead of the geometric label assignment, and the former does not necessitate accurate IoU. In Sec. 5.5, we quantitatively demonstrate the adverse effect of the ambiguity is negligible.

### 4.3 POINTSAM FOR COARSE LABEL GENERATION

The utilization of cluster-level labels has greatly decreased the demand for human annotation. To further reduce the annotation burden of coarse labels, we propose *PointSAM* for automated coarse labeling, resorting to the mighty SAM (Kirillov et al., 2023) to generate coarse cluster-level labels. PointSAM is illustrated in Fig. 5, which comprises two modules: (1) SAM-based 3D Instance Segmentation: We use SAM to infer over-segmented masks and map them to 3D point clouds. (2) Separability-

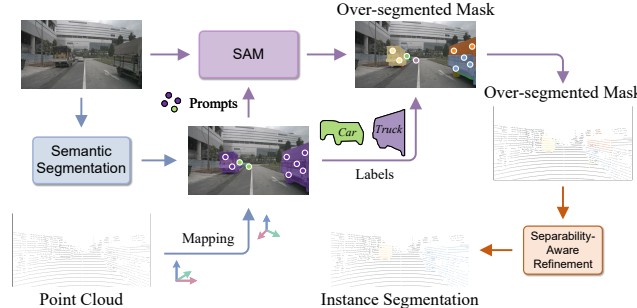

Figure 5: **Overall of PointSAM.**

Aware Refinement: Since SAM's over-segmentation and imprecise point-pixel projection, we propose SAR to mitigate the issues to enhance the quality of segmentation.

**SAM-assisted 3D Instance Segmentation**  We first utilize a pre-trained semantic segmentation model to generate 2D semantic mask. We then project 3D points into the 2D semantic mask. The points mapped into 2D foreground semantic masks serve as prompts for SAM to generate 2D over-segment masks, which significantly improves inference speed. For each mask generated by SAM, the semantic label is assigned based on the category with the highest pixel count within the mask. By the 3D-2D projection, we obtain initial 3D instance masks.

**Separability-Aware Refinement (SAR)**  Nonetheless, the over-segmentation of SAM and projection errors lead to mediocre segmentation quality. For example, there might be some points belonging to the same objects are assigned with different mask IDs or two far apart clusters in the same direction may be assigned the same mask ID. Fortunately, these issues can be alleviated by leveraging the spatial separability property inherent to point clouds. Specifically, we employ connected components labeling (CCL) on the foreground points. After performing CCL, we obtain multiple components. We split those masks which are across multiple components and then merge those masks belonging to a single component. A simple illustration of SAR can be found in Appendix C.2. We explore the resistance of SAR to inaccurate calibration in Appendix C.3. The comparison between PointSAM and other SAM-based methods for 3D tasks is presented in Appendix C.4.

### 4.4 TRAINING LOSS

During training stage, coarse cluster labels only contribute to classification (or confidence) $\mathcal{L}_{cls}$ and accurate box labels only contribute to regression $\mathcal{L}_{reg}$. Based on label assignment, we denote positive samples assigned with accurate labels as $\mathbf{S}_a$, positive samples assigned with coarse labels as $\mathbf{S}_c$, and negative samples as $\mathbf{S}_n$. The loss function for MixSup can be formulated as Eq. 2.

$$\mathcal{L} = \frac{1}{|\mathbf{S}_a \cup \mathbf{S}_c \cup \mathbf{S}_n|} \sum_{\mathbf{S}_a \cup \mathbf{S}_c \cup \mathbf{S}_n} \mathcal{L}_{cls} + \frac{1}{|\mathbf{S}_a|} \sum_{\mathbf{S}_a} \mathcal{L}_{reg}. \tag{2}$$

### 4.5 Discussion: Comparing MixSup with other Label-efficient Methods

MixSup and other label-efficient learning settings such as semi/weakly/self-supervised frameworks serve the same purpose of improving label efficiency. However, they are quite different in terms of design philosophy. For example, weakly supervised methods focus on how to utilize a certain type of weak labels. Popular semi-supervised methods design training schemes such as self-training to generate high-quality pseudo labels. MixSup follows a more practical philosophy to utilize different types of supervision and tries to integrate them into popular detectors for generality.

Thanks to such essential differences, MixSup can seamlessly collaborate with other settings for better performance. To demonstrate the potential, in Sec. 5.5, we establish a simple baseline to utilize the self-training technique brought from semi-supervised learning. We will pursue a more effective combination of MixSup and other label-efficient methods in future work.

## 5 Experiments

### 5.1 Dataset

**nuScenes** nuScenes (Caesar et al., 2020) is a popular dataset for autonomous driving research. It requires 10 object classes, so it is an ideal testbed to evaluate semantic learning with massive coarse labels. Since nuScenes also contains a panoptic segmentation benchmark (Fong et al., 2022), we use it to validate the effectiveness of PointSAM and evaluate the quality.

**Waymo Open Dataset (WOD)** Waymo Open Dataset (Sun et al., 2020) is a widely recognized dataset utilized for 3D object detection. The evaluation metric for WOD is 3D IoU-based mean Average Precision. We set the IoU thresholds of 0.7 for vehicles and 0.5 for pedestrians and cyclists, following official guidelines. Such strict metric makes it a challenging benchmark for MixSup, since it only relies on a limited number of accurate box-level labels for geometry estimation.

**KITTI** KITTI (Geiger et al., 2012) is one of the earliest datasets for 3D detection evaluation. Due to the occlusion and truncation levels of objects, the evaluation is reported with three difficulty levels: easy, moderate, and hard. Here we present the results in terms of the mean Average Precision (mAP) with 11 recall positions under moderate difficulty. IoU thresholds for Car, Pedestrian, and Cyclist are set as 0.7, 0.5, and 0.5, respectively.

### 5.2 Implementation Details

To demonstrate the versatility of our method, we integrate four prominent detectors into MixSup. These include an anchor-based detector SECOND (Yan et al., 2018), an anchor-free detector CenterPoint (Yin et al., 2021), a two-stage detector PV-RCNN (Shi et al., 2020a), and an emerging fully sparse detector FSD (Fan et al., 2022b; 2023b). Notably, SECOND, PV-RCNN, and FSD leverage box-based assignment, while CenterPoint adopts center-based assignment.

We randomly choose 10% and 1% of ground truth boxes to serve as box-level labels. However, in Waymo Open Dataset, since the Cyclist class is very rare, we give it more possibility to be selected. This is essentially another superiority of MixSup: we can flexibly adjust the budget as needed instead of following frame-by-frame selection in conventional methods. These selected labels also function as the database for CopyPaste augmentation, as opposed to the default database copied from fully labeled frames. The implementation of MixSup is based popular codebases MMDetection3D (Contributors, 2020) and OpenPCDet (Team, 2020). The training schedule and hyperparameters are all the same as the fully-supervised training, and all experiments are conducted in 8 RTX 3090 GPUs.

In PointSAM, we solely employ the semantic segmentation head of HTC (Chen et al., 2019), pretrained on nuImages, to obtain semantics. Notably, due to the negligible overlap in image data between nuImages and nuScenes, there is no data leakage during the process of PointSAM.

### 5.3 Labeling Protocol and Cost Analysis

We ask experienced annotators to label 100 frames from different sequences of nuScenes. They follow this protocol: Making three coarse clicks around and object in Bird Eye's View, and the three click points are regarded as three corners of a parallelogram. The points inside the parallelogram form a coarse cluster. We provide a simple illustration in Appendix D.1. The annotators time the whole process. The average time cost of a coarse cluster label is only **14%** of an accurate box

Table 2: Performances on WOD and nuScenes validation split. †: Using coarse cluster labels and 10% accurate box labels. The percentage in parentheses indicates the performance ratio to the fully supervised counterpart.

| Detector | WOD (L2 APH) | | | | nuScenes | |
| | Mean | Vehicle | Pedestrian | Cyclist | mAP | NDS |
|---|---|---|---|---|---|---|
| CenterPoint (100% frames) | 64.66 | 65.04 | 61.20 | 67.73 | 62.41 | 68.20 |
| CenterPoint (10% frames) | 51.64 | 55.39 | 49.07 | 50.46 | 42.19 | 55.38 |
| CenterPoint (MixSup) † | 62.34 (96.41%) | 61.83 (95.06%) | 57.72 (94.31%) | 67.46 (99.60%) | 60.73 (97.31%) | 66.46 (97.45%) |

label. In experiments, we obtain clusters by using noisy GT boxes to crop the inside points. The GT boxes are randomly expanded 0% to 10% in each dimension to mimic the potential noise in coarse labeling. To clearly demonstrate the cost of annotation requirement of MixSup, we unify the annotation costs of box-level and cluster-level labels as Eq. 3.

$$\text{cost} = (N_b + 0.14N_c)/N_t, \tag{3}$$

where $N_b$, $N_c$ represent the number of box-level labels and cluster-level labels, and $N_t$ denotes the total number of labels in training set.

## 5.4 MAIN RESULTS

**Performance on mainstream datasets.** We first showcase the main results of MixSup in Table 2 (WOD and nuScenes), and Table 3 (KITTI). In particular, SECOND and PV-RCNN exhibit performance up to 95.20% of fully-supervised methods, demonstrating the effectiveness of box-based assignment for cluster-level labels. CenterPoint achieves performance levels between 94.31% and 99.60% of fully supervised performance, validating the feasibility of center-based assignment.

Table 3: Performances on KITTI validation split with moderate difficulty. Notations have the same meanings as those in Table 2.

| Detector | Car | Pedestrian | Cyclist |
|---|---|---|---|
| SECOND (100% frames) | 78.62 | 52.98 | 67.15 |
| SECOND (MixSup) † | 74.85 (95.20%) | 50.18 (94.71%) | 61.46 (91.53%) |
| PV-RCNN (100% frames) | 83.61 | 57.90 | 70.47 |
| PV-RCNN (MixSup) † | 76.09 (91.01%) | 54.33 (93.83%) | 65.67 (93.19%) |

**Comparison with other label-efficient frameworks.** Besides MixSup, there are several other label-efficient frameworks such as semi-supervised learning and self-supervised learning. Due to their different settings, an absolute fair comparison cannot be established among these methods. However, in order to gain an intuitive understanding of performance, we list their performances in Table 4, 5. The results suggest that MixSup is an effective paradigm, achieving better or on-par performance, compared with semi/self-supervised settings. We emphasize that MixSup and these methods are complementary and compatible with each other, which will be briefly demonstrated in Sec. 5.5.

Table 4: Comparison with other weakly-supervised learning on KITTI validation split (Car).

| Label-efficient method | Annotation | Easy | Moderate | Hard |
|---|---|---|---|---|
| WS3D (Meng et al., 2020) | 534 boxes + weak labels | 84.04 | 75.10 | 73.29 |
| WS3D (Meng et al., 2021) | 534 boxes + weak labels | 85.04 | 75.94 | **74.38** |
| MixSup (ours) | 534 boxes + weak labels | **86.37** | **76.20** | 72.36 |

## 5.5 PERFORMANCE ANALYSIS

**Comparison with handcrafted box fitting.** Fitting pseudo box-level labels from cluster-level labels, such as L-shape fitting (Zhang et al., 2017), presents a trivial option for incorporating coarse labels into training. However, these methods cannot distinguish length and width, and are also confusing with a certain heading $\theta$ and a heading $\theta + \pi$. Therefore, we ignore the shape and heading supervision during the training. The results in Table 6 manifest box fitting is sub-optimal, particularly for large objects like Car and Truck. This is due to the fact that the point clusters of these large objects are more prone to displaying incomplete object parts. Consequently, the pseudo boxes derived from these clusters exhibit unreliable sizes.

**Integration with simple self-training.** As we discussed in Sec. 4.5, MixSup can collaborate with semi-supervised methods. Deliberately designing the semi-supervised training scheme is out of

Table 5: Comparison with other label-efficient detectors on Waymo Open Dataset validation split (L2 mAPH). †: The annotation cost contains both box labels and cluster labels, defined by Eq. 3. *: From ProficientTeacher (Yin et al., 2022a). §: From MV-JAR (Xu et al., 2023). ¶: From HS-DDA (Liu et al., 2023a).

| Detector | Label-efficient method | Annotation | Mean | Vehicle | Pedestrian | Cyclist |
|---|---|---|---|---|---|---|
| SECOND | - | all frames | 57.23 | 63.33 | 51.31 | 57.05 |
| SECOND* | - | 10% frames | 49.11 | 56.81 | 41.91 | 48.62 |
| SECOND* | FixMatch (Sohn et al., 2020) | 10% frames | 51.45 | 58.37 | 44.23 | 51.75 |
| SECOND* | ProficientTeacher (Yin et al., 2022a) | 10% frames | 54.16 | **59.36** | 46.97 | 56.15 |
| SECOND | MixSup (ours) | 10% annotation cost † | **54.23** | 55.02 | **49.61** | **58.06** |
| SST | - | all frames | 65.54 | 64.56 | 64.89 | 67.17 |
| SST§ | - | 10% frames | 50.46 | 54.37 | 50.71 | 46.29 |
| SST§ | PointContrast (Xie et al., 2020) | 10% frames | 49.94 | 54.30 | 50.12 | 45.39 |
| SST§ | ProposalContrast (Yin et al., 2022b) | 10% frames | 50.13 | 54.71 | 50.39 | 45.28 |
| SST§ | MV-JAR (Xu et al., 2023) | 10% frames | 54.06 | 58.00 | 54.66 | 49.52 |
| SST | MixSup (ours) | 10% annotation cost † | **60.74** | **59.10** | **60.00** | **63.13** |
| PV-RCNN | - | all frames | 67.06 | 68.98 | 64.42 | 67.79 |
| PV-RCNN¶ | - | 1% frames | 20.90 | 43.30 | 15.90 | 2.90 |
| PV-RCNN¶ | HSDDA (Liu et al., 2023a) | 1% frames | 28.27 | 47.30 | 17.50 | 20.00 |
| PV-RCNN | MixSup (ours) | 1% annotation cost † | **56.58** | **55.46** | **52.02** | **62.25** |

Table 6: Comparison with handcrafted box fitting on nuScenes. We adopt CenterPoint as the base detector, conducting training for 10 epochs. †: Ignore the shape and heading supervision for fitted pseudo boxes. ‡: Ignore the heading supervision for pseudo boxes.

| Label Format | mAP | NDS | Car | Truck | C.V. | Bus | Trailer | Bar. | Mot. | Byc. | Ped. | T.C. |
|---|---|---|---|---|---|---|---|---|---|---|---|---|
| MixSup (cluster-level) | 59.48 | 64.97 | 82.35 | 53.65 | 19.17 | 67.25 | 36.87 | 66.48 | 64.79 | 53.45 | 83.46 | 67.29 |
| MixSup (fitted pseudo boxes†) | 55.75 | 62.33 | 63.72 | 45.04 | 19.51 | 65.80 | 21.42 | 67.57 | 65.09 | 56.87 | 84.24 | 68.25 |
| MixSup (fitted pseudo boxes‡) | 56.22 | 60.54 | 65.39 | 44.85 | 20.74 | 67.23 | 25.45 | 63.91 | 66.47 | 56.90 | 83.70 | 67.54 |

the scope of this paper. For simplicity and generality, we establish a *simple self-training* baseline to verify our claim, which is one of the most common techniques in semi-supervised learning.

In particular, we first use a trained MixSup detector to generate pseudo boxes in the training set, and pseudo boxes with scores higher than 0.7 are utilized to replace corresponding coarse cluster labels. Then the updated label set is adopted to train a new detector. As shown in Table 7, the simple self-training strategy consistently improves the performance, indicating MixSup is compatible with semi-supervised training schemes. We will delve into the combination of MixSup and semi-supervised framework in future work.

Table 7: Integration with simple self-training on KITTI validation split with moderate difficulty.

| Detector | Car | Pedestrian | Cyclist |
|---|---|---|---|
| SECOND (100% frames) | 78.62 | 52.98 | 67.15 |
| SECOND (MixSup) | 74.85 | 50.18 | 61.46 |
| Above + self-training | 77.46 | 56.89 | 64.40 |
| PV-RCNN (100% frames) | 83.61 | 57.90 | 70.47 |
| PV-RCNN (MixSup) | 76.09 | 54.33 | 65.67 |
| Above + self-training | 78.87 | 61.03 | 70.91 |

**Roadmap from coarse clusters to accurate boxes.** To better understand the gap and differences between MixSup and fully supervised detectors. We incrementally incorporate additional supervisory information for cluster-level labels. Specifically, we sequentially augment the 90% original cluster-level labels with objects' center coordinates, shape dimensions, and heading step by step. We employ CenterPoint as the fundamental detector and conduct experiments on nuScenens for 10 epochs. The noteworthy enhancements, as detailed in Table 8, are primarily observed in large objects, like Car and Truck. This can be attributed to the fact that the centers of these large-size cluster labels exhibit a more significant deviation from their true box centers.

**The ambiguity of box-cluster IoU.** As mentioned in Sec. 4.2, the proposed box-based assignment relies on box-cluster IoU, which is inherently more ambiguous compared to the IoU between the proposal and the box-level labels. To demystify the effect of such ambiguity, we establish the following *oracle* experiment: Based on FSD, a state-of-the-art two-stage detector, we adopt standard box-to-box IoU for the matching between proposal and GTs during the label assignment of the second stage. The learning scheme after the matching is the same as MixSup, where only 10% of proposals are supervised by accurate poses and shapes.

Table 8: Roadmap from coarse cluster labels to accurate box labels on nuScenes. We adopt Center-Point as the base detector, conducting training for 10 epochs. †: This setting is equivalent to the fully supervised baseline, while its performance is slightly worse due to the shorter training schedule.

| Supervision | mAP | NDS | Car | Truck | C.V. | Bus | Trailer | Bar. | Mot. | Byc. | Ped. | T.C. |
|---|---|---|---|---|---|---|---|---|---|---|---|---|
| MixSup | 59.48 | 64.97 | 82.35 | 53.65 | 19.17 | 67.25 | 36.87 | 66.48 | 64.79 | 53.45 | 83.46 | 67.29 |
| MixSup + Center | 60.49 | 65.89 | 83.85 | 57.00 | 19.66 | 69.09 | 37.98 | 66.30 | 65.29 | 53.86 | 83.84 | 68.00 |
| MixSup + Center + Shape | 60.79 | 66.27 | 83.80 | 56.93 | 21.30 | 70.01 | 37.44 | 67.35 | 64.93 | 54.11 | 83.82 | 68.17 |
| MixSup + Center + Shape + Heading † | 60.95 | 66.31 | 83.79 | 57.20 | 21.75 | 69.09 | 36.99 | 66.90 | 66.96 | 54.24 | 84.12 | 68.44 |

Table 11: Performances with generated labels by PointSAM on nuScenes validation split. *: Using labels from PointSAM. †: Removing false positive clusters. ‡: Adding false negatives based on †.

| Detector | mAP | NDS | Car | Truck | C.V. | Bus | Trailer | Bar. | Mot. | Byc. | Ped. | T.C. |
|---|---|---|---|---|---|---|---|---|---|---|---|---|
| CenterPoint (10% frames) | 42.19 | 55.38 | 77.18 | 38.18 | 3.60 | 42.17 | 9.12 | 59.29 | 36.31 | 20.54 | 78.97 | 56.57 |
| CenterPoint (MixSup)* | 49.49 | 58.65 | 64.63 | 41.71 | 15.61 | 57.57 | 28.19 | 43.56 | 62.28 | 51.42 | 75.07 | 54.87 |
| CenterPoint (MixSup†) | 53.09 | 60.93 | 70.81 | 43.66 | 15.66 | 62.05 | 30.40 | 59.92 | 63.37 | 48.80 | 77.27 | 59.00 |
| CenterPoint (MixSup‡) | 58.30 | 64.21 | 80.33 | 50.74 | 20.59 | 65.38 | 36.11 | 65.52 | 62.45 | 51.92 | 82.06 | 67.89 |

As can be seen from Table 9, there is no significant performance boost in the oracle experiments, demonstrating that MixSup does not necessitate precise IoU measurements. The performance is especially robust for small objects like Pedestrian and Cyclist, indicating the box-cluster IoU is sufficient in semantics learning even if it is a little ambiguous.

Table 9: Oracle study to understand the ambiguity in box-based assignment, on Waymo Open Dataset validation split (L2 mAPH).

| IoU type | Mean | Vehicle | Pedestrian | Cyclist |
|---|---|---|---|---|
| cluster-to-box | 68.57 | 66.08 | 66.53 | 73.09 |
| box-to-box (oracle) | 68.96 | 67.17 | 66.75 | 72.95 |

## 5.6 ANALYSIS OF POINTSAM

**Quantitative Analysis** We perform PointSAM for automated coarse labeling on nuScenes and compare the labels with prior arts on LiDAR-based panoptic segmentation benchmark (Fong et al., 2022). As PointSAM disregards background, we only report the performance for foreground *thing* classes, in Table 10. Thanks to the mighty SAM, PointSAM is on par with the recent fully supervised panoptic segmentation models without any 3D annotations.

Table 10: Panoptic segmentation performance for *thing* classes on nuScenes validation split.

| Methods | PQ | SQ | RQ |
|---|---|---|---|
| GP-S3Net (Razani et al., 2021) | 56.0 | 85.3 | 65.2 |
| SMAC-Seg (Li et al., 2022) | 65.2 | 87.1 | 74.2 |
| Panoptic-PolarNet (Zhou et al., 2021) | 59.2 | 84.1 | 70.3 |
| SCAN (Xu et al., 2022) | 60.6 | 85.7 | 70.2 |
| CFNet (Li et al., 2023) | 74.8 | 89.8 | 82.9 |
| **PointSAM (Ours)** | **63.7** | **82.6** | **76.9** |

**Human Rectification** Although SAM usually generates high-quality clusters, there are inevitable false-positive clusters and false negatives due to the errors of 3D-2D projection in nuScenes. These errors cannot be completely fixed due to imprecise calibration of sensors. We provide the analysis of these bad cases in Appendix C.1. Thus, we manually correct the false positive labels and false negatives, according to the labeling protocol in Sec. 4.1. The human rectification leads to significant results in Table 11, at a cost of 50% annotation burden of all coarse labels.

## 6 CONCLUSION AND FUTURE WORK

Based on the unique properties of point clouds, we first verify that a good LiDAR-based detector needs massive coarse labels for semantic learning but only a few accurate labels for geometry estimation. We then propose a general label-efficient LiDAR-based framework MixSup to utilize massive cheap cluster labels and a few accurate box labels. In addition, we develop PointSAM to further reduce the annotation burden. The effectiveness is validated in three mainstream benchmarks.

MixSup has great potential to collaborate with well-studied semi-supervised methods. We have shown the potential with a simple attempt and will delve into the relevant investigation in the future. Moreover, the emerging auto-labeling methods, such as (Yang et al., 2021; Qi et al., 2021; Fan et al., 2023a; Ma et al., 2023), present a compelling way to generate massive coarse labels. These automatic labelers can be utilized to further improve the performance of MixSup.

ACKNOWLEDGMENTS

This work was supported in part by the National Key R&D Program of China (No.2022ZD0116500), the National Natural Science Foundation of China (No.U21B2042, No.62320106010, No.62072457), and in part by the 2035 Innovation Program of CAS.

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

## A    PILOT STUDY

### A.1    WELL-CLASSIFIED DATASET

As discussed in Sec. 3, we showcase some samples from the well-classified dataset in Fig. 6. In the study, we utilize a pre-trained FSD (Fan et al., 2022b; 2023b) for region cropping. Each region is deliberately introduced with strong noise against leakage of geometry information.

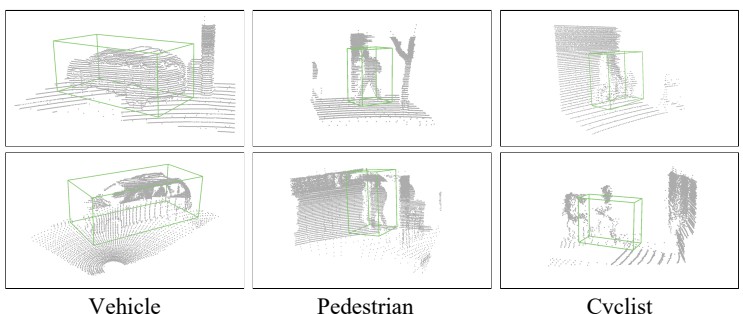

Figure 6: **Samples from the well-classified dataset.** Green bounding boxes represent the ground truth, while the presence of abundant background points underscores the strong noise we have introduced.

### A.2    EXPLORATION ON SEMANTIC LEARNING IN PILOT STUDY

We conduct an experiment to explore the impact of different data amounts on the 3D detector's semantic learning. Specifically, we decrease the data amount (the number of training frames) from 100% to 10% in Waymo and train a popular detector CenterPoint (Yin et al., 2021). Such a decrease in data amount has a negative impact on both geometry learning and semantic learning. To mitigate the impact on geometry learning and only focus on semantic learning, we aggressively relax the IoU thresholds in evaluation from 0.7 / 0.5 / 0.5 to 0.5 / 0.25 / 0.25 for Vehicle / Pedestrian / Cyclist to mitigate the negative impact introduced by degradation of geometry estimation.

Table 12: Performances with varying data amounts and IoU thresholds on Waymo Open Dataset validation split.

| Data amount | IoU thresholds | Vehicle L2 AP / APH | Pedestrian L2 AP / APH | Cyclist L2 AP / APH |
|---|---|---|---|---|
| 100% | 0.7 / 0.5 / 0.5 (normal) | 65.42 / 64.92 | 66.49 / 60.53 | 66.49 / 60.53 |
| 10% | 0.7 / 0.5 / 0.5 (normal) | 55.92 / 55.39 | 56.97 / 49.07 | 51.52 / 50.46 |
| Performance Gap | - | 9.50 / 9.53 | 9.52 / 11.46 | 17.76 / 17.66 |
| 100% | 0.5 / 0.25 / 0.25 (relaxed) | 87.15 / 86.20 | 82.99 / 74.62 | 74.01 / 72.70 |
| 10% | 0.5 / 0.25 / 0.25 (relaxed) | 81.99 / 80.67 | 73.61 / 62.14 | 56.50 / 55.20 |
| Performance Gap | - | 5.16 / 5.53 | 9.38 / 12.48 | 17.51 / 17.50 |

As shown in Table 12, there are huge performance gaps between using 100% data and using 10% data under the normal IoU thresholds. Obviously, here these gaps are caused by the degradation of **both** semantic learning and geometry estimation.

However, there are still significant gaps between using 100% data and using 10% data under the relaxed IoU thresholds. Especially, for Pedestrian and Cyclist, the gaps (between 10% and 100%) do not even get smaller after we relax the IoU thresholds. Thus, the performance gaps between 100% data and 10% data should be mainly caused by the degradation of semantic learning. In other words, semantic learning is sensitive to the data amount.

From another point of view, in the pilot study in Sec. 3, we have decreased the data amount of well-classified patches from 100% to 10% to reveal the impact on geometry estimation, as shown in Table 1. Compared with the aforementioned performance change caused by semantic learning degradation, the performance change in Table 1 is negligible. Thus, we draw the conclusion that

"LiDAR-based detectors indeed only need a very limited number of accurate labels for geometry estimation. Massive data is only necessary for semantic learning".

## B  Discussion on Weakly-supervised Learning

Since there is limited research on weakly-supervised learning for LiDAR-based 3D object detection, we take WS3D (Meng et al., 2020; 2021), one of the closest work to MixSup, as an example for discussion and comparison, while the direct comparison on the performance is shown in Table 4.

In terms of implementation, WS3D has a specifically designed detection pipeline and cannot be generalized to other detectors. In contrast, MixSup can be integrated with various detectors. Additionally, the coarse cluster-level labels proposed in Sec. 4.1 can be obtained through foundational models such as SAM, as we demonstrated in PointSAM (Sec.4.3), further reducing the annotation burden.

From the perspective of labeling protocol, WS3D requires annotators to click object centers in the BEV perspective, which are used to supervise the center prediction. However, we demonstrate that such center-click labeling protocol has some limitations as follows.

- For the inevitable instances with very partial point clouds, accurately clicking the center is challenging. Conversely, MixSup's cluster-level label only requires clicking around the visible part of an object with three points, ensuring that the generated parallelogram encompasses the partial point clouds.

- Some methods need more than centers, such as the recent state-of-the-art FSD (Fan et al., 2022b). It adopts a segmentation network in the first stage to obtain foreground points, requiring point-level supervision. To clearly show the effectiveness of MixSup, we complement the performance on Waymo validation split with L2 APH using FSD as the base detector in Table 13.

Table 13: Performances on WOD validation split. †: Using coarse cluster labels and 10% accurate box labels. The percentage in parentheses indicates the performance ratio to the fully supervised counterpart.

| Detector | WOD (L2 APH) | | | |
| --- | --- | --- | --- | --- |
| | *Mean* | *Vehicle* | *Pedestrian* | *Cyclist* |
| FSD (100% frames) | 71.27 | 70.09 | 69.79 | 73.93 |
| FSD (MixSup) † | 68.57 (96.21%) | 66.08 (94.28%) | 66.53 (95.33%) | 73.09 (98.86%) |

## C  PointSAM

### C.1  Qualitative Analysis

We list the visualization of generated coarse labels in Fig. 7. As depicted in subfigures (a) and (b), PointSAM demonstrates the capability to generate high-quality cluster-level labels. However, due to low-quality segmentation in extreme situations or 3D-2D projection errors, it is inevitable to arise false-positive clusters and false negatives, as shown in subfigures (c) and (d). In particular, subfigure (c) showcases an extreme case of nighttime driving, where SAM fails to provide effective segmentation masks, resulting in untrustworthy cluster labels. Subfigure (d) exemplifies a case of incorrect projection, where background points are erroneously projected to the foreground labels.

### C.2  Separability-Aware Refinement (SAR)

A simple illustration of SAR is shown in Fig. 8. For those masks across multiple components, we initially analyze each mask's number in each component and retain the one with the highest count. It ensures that each mask is associated with only one component. Subsequently, we merge the masks that belong to a single component and output the final segmentation masks.

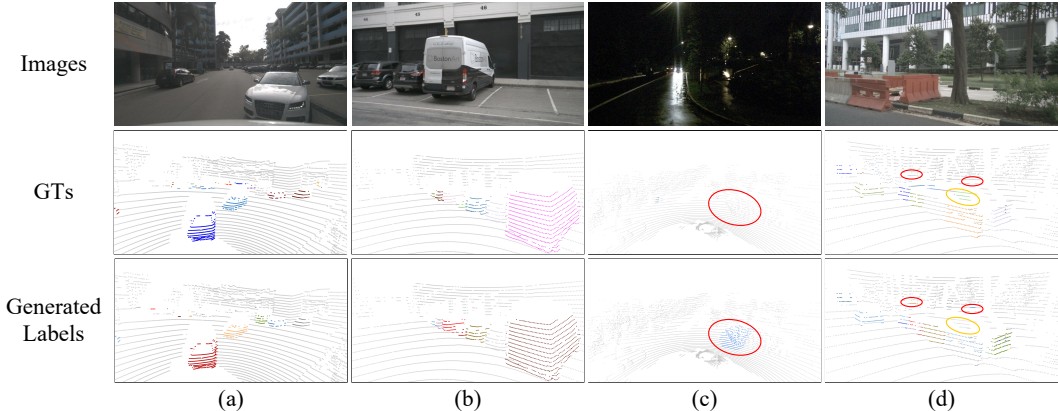

Figure 7: **Visualization of generated labels through PointSAM.** Subfigures (a) and (b) depict accurately generated samples, while subfigures (c) and (d) illustrate samples containing false-positive clusters in red circles and false negatives in yellow circles.

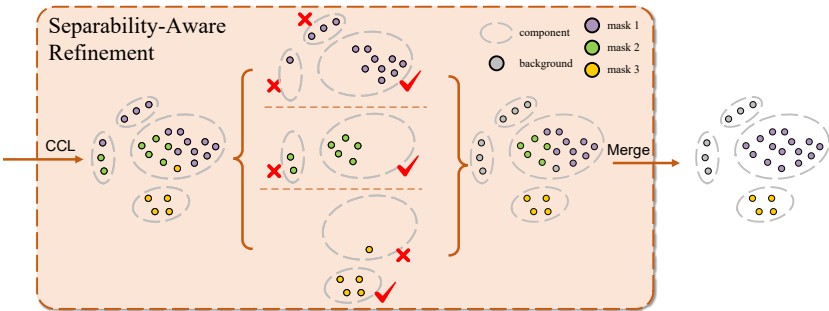

Figure 8: **Illustration of Separability-Aware Refinement (SAR).** In SAR, ✓ denotes retaining the mask and ✗ denotes to regard these points as background.

### C.3 RESISTANCE OF SAR TO INACCURATE CALIBRATION

Throughout the experiment, we identified inherent inaccuracies in the calibration of nuScenes, leading to discrepancies in the pixel-point projection of foreground objects. Hereby, we introduce SAR module to mitigate the degradation in segmentation caused by projection error.

To further investigate the impact of projection discrepancies on PointSAM, we introduced random noise to each camera's position. The performance for foreground objects on nuScenes val split panoptic segmentation is summarized in Table 14. The results indicate that SAR-derived coarse instance masks exhibit a certain degree of resistance to calibration inaccuracies, attributed to the refinement introduced by the SAR module.

Table 14: Performances with noisy calibration.

| Noise (cm) | w/ SAR | PQ | SQ | RQ |
|---|---|---|---|---|
| 0 | ✗ | 44.9 | 74.6 | 59.8 |
| $-5 \sim 5$ | ✗ | 43.8 | 74.6 | 58.3 |
| $-10 \sim 10$ | ✗ | 41.4 | 74.5 | 55.2 |
| 0 | ✓ | 63.7 | 82.6 | 76.9 |
| $-5 \sim 5$ | ✓ | 62.8 | 82.3 | 76.1 |
| $-10 \sim 10$ | ✓ | 60.6 | 81.4 | 74.3 |

### C.4 DISCUSSION ON SAM-BASED METHODS

To the best of our knowledge, PointSAM is the first initiative harnessing SAM for instance segmentation in outdoor scenes. Notably, PointSAM achieves performance on par with the recent fully supervised panoptic segmentation models without the need for any 3D annotations as shown in Table 10. Moreover, our PointSAM innovatively leverages the inherent spatial separability of point clouds to refine instance segmentation, enabling the mitigation of the projection error.

In terms of other SAM-based works, taking SAM3D for detection (Zhang et al., 2023a), SAM3D for instance segmentation (Yang et al., 2023b), Seal (Liu et al., 2023c), and Label-free Scene Understanding (Chen et al., 2023a) as examples, SAM3D for detection (Yang et al., 2023a) is an early

exploratory work that innovatively applies SAM to LiDAR-based 3D object detection. However, it has some limitations, such as being confined to detecting only vehicles and impractical performance. SAM3D for instance segmentation (Yang et al., 2023b) focuses on multi-view merging in indoor scenes, while PointSAM pays attention to refining based on the spatial separability in outdoor scenes. Moreover, we integrate semantics to enrich the information for instance masks. The latter two works (Liu et al., 2023c; Chen et al., 2023a) both focus on pretraining via pixel-point projection for semantic segmentation and have achieved remarkable results. However, they cannot handle instance segmentation.

It is crucial to emphasize that PointSAM is proposed to provide coarse labels for MixSup. Thus, we are delighted to see these exceptional works playing a similar role and potentially integrating with MixSup.

## D  MANUALLY LABELING

### D.1  LABELING PROTOCOL

We outline the labeling protocol for cluster-level labels in Fig. 9. As described in Sec. 4.1, we can annotate a cluster with just three clicks, which is significantly more efficient compared to annotating box-level labels.

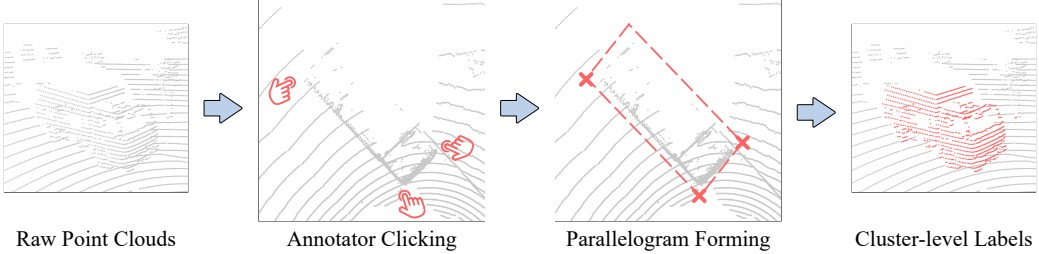

Raw Point Clouds     Annotator Clicking     Parallelogram Forming     Cluster-level Labels

Figure 9: **Illustration of labeling protocol for cluster-level labels.** Note the annotator clicking can be very fast and they do not need to click the exact corners.

### D.2  NOISE IN HUMAN-ANNOTATED COARSE LABEL

To delve into the impact of the noise introduced by annotations on our experiments, we first evaluate the similarity between human-annotated coarse labels (100-frame subset) with ground-truth labels, **which is 81.22% in terms of segmentation mIoU**. Then we add stronger noise to ground-truth labels to simulate more rigorous coarse labels, controlling coarse labels possessing **80% to 82%** mIoU with ground-truth labels, just like human-annotated coarse labels. In particular, we use the following two noises:

- Noise ①: Randomly shift center locations $-0.1 \sim 0.1$ meters and expand $0\% \sim 50\%$ in each dimension. It possesses $81.91\%$ mIoU with ground-truth labels.
- Noise ②: Randomly shift center locations $-0.2 \sim 0.2$ meters, expand $0\% \sim 20\%$ in each dimension, and rotate $-15° \sim 15°$ in heading. It possesses $80.60\%$ mIoU with ground-truth label.

We then conduct experiments on nuScenes using such noisy coarse labels. As shown in Table 15, performance with noise ①, ② is similar to the performance with default noise ⓪. Thus, our experiments conducted with default noise are reliable and won't introduce significant deviations from the manually annotated coarse labels.

## E  LIMITATIONS AND SOCIAL IMPACT

Regarding the limitations of MixSup, as a novel label-efficient paradigm, it is orthogonal to other label-efficient methods. Our work has not explored integration with semi-supervised methods, pro-

Table 15: Comparison with different noisy coarse labels on nuScenes. Noise ⓪: Default noise introduced by randomly expanding $0\% \sim 10\%$ in each dimension.

| Detector | Noise | mAP | NDS |
|---|---|---|---|
| CenterPoint (100%) | - | 62.41 | 68.20 |
| CenterPoint (MixSup) | Noise ⓪ | 60.73 | 66.46 |
| CenterPoint (MixSup) | Noise ① | 60.23 | 65.99 |
| CenterPoint (MixSup) | Noise ② | 60.21 | 66.28 |

viding an avenue for potential performance enhancements. Moreover, our proposed PointSAM, when combined with other exceptional 3D segmentation models, could yield higher-quality coarse labels. We believe MixSup holds significant promise in reducing annotation costs for the community, contributing to the conservation of both human and environmental resources. However, due to the cost savings in annotation come with a certain performance trade-off, practical deployment may raise risks of compromising driving safety. Thus, these will be a focus of our future research.

