# OpenReview forum: "MixSup: Mixed-grained Supervision for Label-efficient LiDAR-based 3D Object Detection"
_ICLR.cc/2024/Conference — ICLR 2024 poster_

### Official Review · Reviewer_unqt · 2023-10-26

**Soundness:** 3 good
**Presentation:** 3 good
**Contribution:** 3 good
**Rating:** 6
**Confidence:** 4

**Summary:**

This paper introduce MixSup, a 3D object detection model that allow for mixed-grained supervision. The paper describe the phenomenon that a general point clouds detector rely heavily on coarse label for semantic recognition and require only few precise labels for object geometry. The method has been adapted for mainstream detectors and tested on various datasets, achieving nearly 97.31% of full supervised performance with reduced annotation.

**Strengths:**

The author's illustrations are well-done, and the article is written in a relatively coherent manner.

**Weaknesses:**

1. The illustration of distinct properties of point clouds compared to images in the introduction section seems not very relevant to the motivation and the concept of the study (The paper is not about multi-modality fusion).
2. The detail of pilot study is not quite clear. It seems that well-classified dataset contains only region-level point cloud. To let a spconv-based detector train on this dataset is not same as training on the point cloud scene, as the scene level point cloud can provide more context information.
3. The motivation "good detector needs massive semantic labels
for difficult semantic learning but only a few accurate labels for geometry estimationhis" is simply another way of expressing a common phenomenon in object detection, which is the high recall often contribute more on high average precision (AP).
4. The author proposed to create cluster-level supervision as the parallelogram of three click. If one can generate massive pseudo-boxes from clicks. Why not just using a region-based network (e.g. the second stage PointNet in LiDAR-RCNN) to refine these boxes as precise box-level supervision?
5. From author's experiment in Table 4, previous methods like MV-JAR can achieve very close performance to author's with 10% anotations. Considering author use massive cluster-level semantic supervision, does this indicate the massive semantic label is not that important?

**Questions:**

1. What is the difference between the settings of 10% boxes annotation and 10% frames annotation?
2. The author use cluster center to substitue the object center for semantic learning. I think this will introduce instability of training due to the cluster containing noise as it is generated from clicking/SAM model. I suggest author can perform another experiment by training the semantic head by leveraging voxel-to-point interpolation (eg. used in PVCNN, SASSD) and imposing point-level supervision.

---

> ### Author Response · Authors · 2023-11-16
> **Response to Reviewer unqt (Part 1)**
>
> We sincerely appreciate your generous suggestions and the thoughtful questions you've provided. We highly value your feedback and will provide detailed explanations to address your concerns.
>
> > W1: The illustration seems not very relevant to the motivation and the concept of the study.
>
> We sincerely apologize for any potential confusion caused by the introduction. In the upcoming revised version, we will present it in a clearer and more precise manner. It's important to clarify that the images in the introduction are intended to offer readers a more vivid and intuitive depiction of the distinctive properties of point clouds.
>
> > W2: The detail of pilot study is not quite clear.
>
> Firstly, we sincerely apologize for any lack of clarity in the pilot study. We indeed trained an spconv-based detector on well-classified regions using varying amounts of data to validate our claim. The purpose of using well-classified regions for training is to factor out the impact on semantic learning, while only focusing on the impact on geometry estimation. Certainly, as you rightfully noted, compared to scene-level point clouds, region-level data does contain less contextual information. However, we believe such a difference does not weaken our conclusion in the pilot study for the following two reasons.
>
> - **Geometry estimation (regression) does not necessitate too much spatial context.** The second stage detector LiDAR-RCNN [R1], employs region-level point clouds to predict accurate object poses. It only incorporates context by enlarging the proposal dimension by **1.5 meters on each side.** Its success demonstrates that precise pose estimation does not necessitate an excessive amount of contextual information. Other two-stage detectors, such as  FSD[R2], also leverage points inside proposals to make accurate pose refinement with a **small proposal enlargement (e.g., 0.5m).** Therefore, our experiments based on region-level data to assess the impact of data amount on regression performance are reasonable.
> - **Our region-based network also utilizes necessary context information.** We adopted a proposal enlargement strategy similar to LiDAR-RCNN, providing sufficient contextual information for regression.
>
> > W3: The motivation is simply another way of expressing a common phenomenon in object detection, which is the high recall often contribute more on high average precision (AP).
>
> We acknowledge that this phenomenon is reasonable in 2D object detection, and we truly thank the reviewer for pointing it out. But it's crucial to note that 3D detection is more sensitive to regression quality compared to its 2D counterpart, due to the following two reasons:
>
> - In the 3-dimensional space, the additional spatial dimension makes achieving high 3D IoU more challenging than achieving high 2D IoU.
> - 3D object detection requires the consideration of orientation angles, which introduces an additional freedom in the matching process between predictions and GTs.
>
> Taking the widely used autonomous driving dataset Waymo as an example, Waymo sets the official IoU threshold as 0.7 for vehicle, and proposes the metric APH that incorporates heading as a weighted factor, which is a strict metric for evaluating a 3D object detector.
>
> To vividly illustrate the rigorous evaluation of geometry attributes in 3D object detection, we conduct experiments with a pre-trained CenterPoint model. Specifically, we introduce different noises to the predicted boxes.
>
> | Noise in location (m) | Noise in dimension (m) | Noise in heading (degree) | Vehicle L1 AP | Vehicle L1 APH | Vehicle L2 AP | Vehicle L2 APH |
> |---|---|---|:---:|:---:|:---:|:---:|
> | - | - | - | 73.43 | 72.88 | 65.42 | 64.92 |
> | 0.05 | 0.05 | 5 | 61.39 | 59.62 | 54.13 | 52.57 |
> | 0.1 | 0.1 | 7.5 | 23.82 | 22.84 | 20.66 | 19.81 |
>
> The introduced noise (e.g., 0.05m) is minor because vehicles are usually more than 4 meters long.
> Experimental results demonstrate that even minor geometry attribute deviations can lead to significant performance drops under the strict metric. This highlights the need for a good 3D detector with strong geometry estimation capabilities, as high recall alone may not guarantee good performance.

---

> ### Author Response · Authors · 2023-11-16
> **Response to Reviewer unqt (Part 2)**
>
> > W4: Why not just using a region-based network (e.g. the second stage PointNet in LiDAR-RCNN) to refine these boxes as precise box-level supervision?
>
> We greatly appreciate your insightful suggestion. It is an insightful suggestion, but there are some potential drawbacks:
>
> - **Pretrained region-based detectors cannot be directly utilized.** There are domain gaps between the human-labeled cluster labels and the input regions of the region-based detector. For example, the input of LiDAR-RCNN comes from the output proposals of other detectors with being enlarged, which is quite different from our cluster labels. So it is infeasible to directly utilize an off-the-shelf pretrained region-based detector. Moreover, the diverse data distributions of different datasets make it more infeasible to use pretrained region-based detectors from other datasets. How to improve the transferring performance of region-based detectors remains open.
>
> - **Transforming coarse labels to precise boxes for supervision introduces inherent errors, reducing the performance upper bound.**  This is because the predicted precise boxes cannot be as good as human annotations. In contrast, although our coarse cluster labels do not have accurate geometry information, the semantic label for each cluster remains precise. Furthermore, the regression learning in MixSup only relies on accurate box-level labels. Thus, although we adopt coarse labels, the supervision in our method does not have such inherent errors and potentially offers a higher performance upper bound.
>
> Although there are these drawbacks, your suggestion is indeed very insightful. We will try to make it more feasible in the future, for example by improving the transferability of region-based detectors.
>
> > W5: MV-JAR can achieve very close performance to author's with 10% annotations.
>
> Thank you for expressing your concerns, and we appreciate the opportunity to clarify this point. It's essential to note that in Table 4, MixSup adopts SECOND as the base detector rather than SST, which is a better detector than SECOND. Therefore, the marginal improvement of MixSup over MV-JAR in Table 4 is reasonable.
>
> To further address your concerns, we also take SST as the base detector to compare with MV-JAR fairly. As shown in the following table, MixSup outperforms MV-JAR with the same base detector, showing that using extra coarse labels leads to improvement.
>
> | Detector | Label-efficient method | Annotation | Mean | Vehicle | Pedestrian | cyclist |
> |---|---|:---:|:---:|:---:|:---:|:---:|
> | SST | MV-JAR | 10% annotation cost | 54.06 | 58.00 | 54.66 | 49.52 |
> | SECOND | MixSup | 10% annotation cost | 54.23 | 55.02 | 49.61 | 58.06 |
> | SST | MixSup | 10% annotation cost | **60.74** | **59.10** | **60.00** | **63.13** |
>
> > Q1: What is the difference between the settings of 10% boxes annotation and 10% frames annotation?
>
> “10% frames annotation” indicates using 10% of all annotated frames, where all objects in each used frame are labeled. “10% boxes annotation" means that the annotated boxes account for 10% of all boxes. It is noteworthy that, due to random sampling and the large scale of driving datasets (e.g., Waymo contains millions of annotated objects), 10% boxes annotation and 10% frames annotation involve almost the same annotation cost.
>
> > Q2: I suggest author can perform another experiment by training the semantic head by leveraging voxel-to-point interpolation (eg. used in PVCNN, SASSD) and imposing point-level supervision.
>
> We sincerely appreciate your insightful suggestion. Your proposal is indeed quite reasonable.
>
> In fact, our method has already employed such supervision in certain situations, such as PV-RCNN [R5]  (tested on KITTI validation split with moderate difficulty). Specifically, PV-RCNN generates features for key points through the voxel set abstraction module, which is very similar to voxel-to-point interpolation, as you pointed out, from neighboring voxels.
>
> Moreover, we supplement a point-based method FSD [R2], which adopts a segmentation network in the first stage to obtain foreground points with **point-level supervision** and test it on Waymo validation split with L2 APH.
>
> Indeed, as you pointed out, MixSup incorporating point-level supervision has shown excellent performance. We sincerely appreciate your valuable suggestion!
>
> | Detector | Car | Pedestrian | Cyclist |
> |---|---|---|---|
> | PV-RCNN (100% frames) | 83.61 | 57.90 | 70.47 |
> | PV-RCNN (MixSup)* | 76.09 (91.01%) | 54.33 (93.83%) | 65.67 (93.19%) |
>
> | Detector | Mean | Vehicle | Pedestrian | Cyclist |
> |---|---|---|---|---|
> | FSD (100% frames) | 71.27 | 70.09 | 69.79 | 73.93 |
> | FSD (MixSup)* | 68.57 (96.21%) | 66.08 (94.28%) | 66.53 (95.33%) | 73.09 (98.86%) |
>
> *: Using **coarse cluster-level labels and 10% box-level labels**. The percentage in parentheses indicates the performance ratio to the fully supervised counterpart

---

> > ### Comment · Reviewer_unqt · 2023-11-19
> > **Follow up on Q1**
> >
> > If 10% boxes annotation and 10% frames annotation involve almost the same annotation cost, is the comparison unfair since the cluster labels are involved?

---

> ### Author Response · Authors · 2023-11-16
> **Response to Reviewer unqt (Part 3)**
>
> ### **References**
>
> - [R1] Li, Zhichao, et al. "Lidar r-cnn: An efficient and universal 3d object detector." *Proceedings of the IEEE/CVF Conference on Computer Vision and Pattern Recognition*. 2021.
> - [R2] Fan, Lue, et al. "Fully sparse 3d object detection." *Advances in Neural Information Processing Systems* 35 (2022): 351-363.
> - [R3] Meng, Qinghao, et al. "Weakly supervised 3d object detection from lidar point cloud." *European Conference on computer vision*. Cham: Springer International Publishing, 2020.
> - [R4] Meng, Qinghao, et al. "Towards a weakly supervised framework for 3d point cloud object detection and annotation." *IEEE Transactions on Pattern Analysis and Machine Intelligence* 44.8 (2021): 4454-4468.
> - [R5] Shi, Shaoshuai, et al. "Pv-rcnn: Point-voxel feature set abstraction for 3d object detection." *Proceedings of the IEEE/CVF conference on computer vision and pattern recognition*. 2020.

---

> ### Author Response · Authors · 2023-11-19
> **Reopens to the Follow-up question**
>
> Thanks for your follow-up question, and truly sorry for causing such a misunderstanding. We clarify our settings and answer this question in the following aspects.
>
> - We list some results in the following table, where the experiments adopt the same "annotation cost". Here the “annotation cost” is a comprehensive metric taking both accurate box annotation and coarse cluster annotation into account. Its details are also listed below.
>
> | Detector | Label-efficient method | Annotation | Mean L2 APH (Waymo val) |
> |---|---|---|:---:|
> | SECOND | - | all frames | 57.23 |
> | SECOND | - | 10% frames | 49.11 |
> | SECOND | FixMatch [R6] | 10% frames | 51.45 |
> | SECOND | ProficientTeacher [R7] | 10% frames | 54.16 |
> | **SECOND** | **MixSup (ours)** | **10% annotation cost** | **54.23** |
> | SST | - | all frames | 65.54 |
> | SST | - | 10% frames | 50.46 |
> | SST | MV-JAR [R8] | 10% frames | 54.06 |
> | **SST** | **MixSup (ours)** | **10% annotation cost** | **60.74** |
>
> - We further explain the details of “10% annotation cost”, which consists of the cost of box labels and coarse cluster labels. The following table shows the statistics.
>
> | #. used box labels | #. used cluster labels | #. all instances in training set |
> |:---:|:---:|:---:|
> | 312,269 | 2,898,813 | 7,061,433 |
>
> As illustrated in Sec 5.3, we conducted a human labeling study by asking three experienced annotators to label 100 frames, containing thousands of objects. Based on the statistics gathered from the annotators, the average time cost of a coarse cluster label is 14% of an accurate box label. In this case, the overall annotation cost is evaluated using the following equation.
>
> $$
> \begin{equation}
> \begin{aligned}
> cost=&\frac{(312269+2898813\times0.14)}{7061433}\times100\\% \\\\
> \approx&4.42\\% + 5.75\\%\\\\
> \approx&10.17\\%
> \end{aligned}
> \end{equation}
> $$
>
>
> Thus, at the same annotation cost, the MixSup also shows superiority.
>
> - Finally, we would like to clarify that there are two reasons why we put “10% boxes annotations + coarse labels” on the table in our paper.
>     - The first reason is that we want to imply that the proposed MixSup could utilize “additional” coarse labels, which is the core feature of MixSup.
>     - The second reason is that we would like to offer readers a more comprehensive understanding of our performance.
>
> **We hope our explanation could resolve your concern. If not, we are open to any further discussion. If it does, we would greatly appreciate it if you could kindly offer us a positive rating. Many thanks in advance.**
>
> ### **References**
>
>
> [R6] Sohn, Kihyuk, et al. "Fixmatch: Simplifying semi-supervised learning with consistency and confidence." *Advances in neural information processing systems* 33 (2020): 596-608.
>
> [R7] Yin, Junbo, et al. "Semi-supervised 3D object detection with proficient teachers." *European Conference on Computer Vision*. Cham: Springer Nature Switzerland, 2022.
>
> [R8] Xu, Runsen, et al. "MV-JAR: Masked Voxel Jigsaw and Reconstruction for LiDAR-Based Self-Supervised Pre-Training." *Proceedings of the IEEE/CVF Conference on Computer Vision and Pattern Recognition*. 2023.

---

> > ### Comment · Reviewer_unqt · 2023-11-22
> >
> > After carefully reading the author's rebuttal and the issues raised by other reviewers, I am inclined to raise my score to 5. My concern still lies in the lack of clarity in the pilot study and experimental settings. I hope the author can incorporate the additional experiments from the rebuttal into the revised manuscript.

---

> > > ### Author Response · Authors · 2023-11-22
> > >
> > > Thank you so much for the feedback and rating. We have provided clarifications about the pilot study and experiments in the **uploaded revision**, including but not limited to
> > >
> > > - A more in-depth analysis of the pilot study and more accurate expressions. [Section 3: Pilot Study; Appendix A.2 Exploration on Semantic Learning in Pilot Study]
> > > - Quantitative comparison with more methods. [Section 5.4 Main Results, Table 4; Appendix B Discussion on Weakly-supervised Learning]
> > > - Quantitative comparison under the exact same annotation cost. [Section 5.4 Main Results, Table 5]
> > > - Experiments with human-level coarse labels. [Appendix D.2 Noise in Human-annotated Coarse Label]
> > > - Robustness to inaccurate calibration in automated coarse labeling. [Appendix C.3 Resistance of SAR to Inaccurate Calibration]
> > >
> > > Thus, we would like to ask **if you could kindly let us know your specific concerns** on “pilot study and experimental settings”? We will definitely try our best to address them. \
> > > Moreover, we promise to release the full code containing the detailed experiment settings.

---

### Official Review · Reviewer_joqH · 2023-10-28

**Soundness:** 3 good
**Presentation:** 3 good
**Contribution:** 3 good
**Rating:** 8
**Confidence:** 4

**Summary:**

This work aims to improve the 3D object detection accuracy using cheap annotations. The authors are motivated by three distinct properties of LiDAR point clouds when designing their method, i.e., texture absence, scale invariance, and geometric richness. The following assumption is then made: *“A good detector needs massive semantic labels for difficult semantic learning but only a few accurate labels for geometry estimation”*.

In their endeavor to substantiate this assumption, an insightful pilot study is put forth. The study's essence is to underscore that extant 3D object detectors excel at deducing geometric information. This is highlighted by the observation that *“detector performance remains relatively stable across data sizes ranging from a mere 5% to a comprehensive 100%”*.

This work then proposes MixSup for conducting mixed-grained supervision on LiDAR-based 3D object detectors. MixSup utilizes massive cheaper cluster labels and a few more accurate box labels for label-efficient learning, where the cluster labels can be retrieved via three coarse clicks around the object corners. To further reduce human annotation effort, the authors use SAM to generate coarse instance labels from the camera images and then refine and map these labels to the point cloud.

Experiments on nuScenes, Waymo Open, and KITTI datasets verified the effectiveness of the proposed approach.

**Strengths:**

- This research is firmly rooted in a well-articulated motivation and logical progression.
- The proposed approach is intuitive and efficient in handling LiDAR-based 3D object detection.
- This paper is overall very well written and pleasant to read. Meanwhile, the authors made a promise of making their code open-source, which could facilitate future research in this task.

**Weaknesses:**

- Several design decisions appear to be largely heuristic in nature. A number of assertions lack rigorous justification or analytical backing.
- The paper omits certain pivotal implementation specifics. This omission hampers a clear assessment of the proposed method's true efficacy.

**Questions:**

### Questions to the Authors

- **Q1:** The pilot study's design and outcomes could gain depth with a more granular breakdown of the experimental setup and subsequent insights. Furthermore, could you elucidate the term “data size”? Is it suggestive of the detectors' adaptability to varying “object/instance sizes”?

- **Q2:** The mechanics of the “clicking annotation” remain somewhat elusive. How were the annotators guided in executing this task? Was there a specific strategy adopted for different instance types? Was this manual labeling extended to all point clouds during training? If not, how were the crucial clicking points discerned? This reviewer believes a more thorough exposition on this subject would be beneficial in the rebuttal phase.

- **Q3:** The acronym “SAR” makes its debut in Sec. 4.3 without any contextual introduction. The authors are suggested to revise this.

- **Q4:** The narrative on connected components labeling (CCL) in Sec. 4.3 is quite succinct, leaving readers wanting more details on its practical implementation.

- **Q5:** It remains unclear which pre-trained semantic segmentation model was harnessed to create the 2D semantic masks. The possible implications of such a model leading to inadvertent data leakage need to be addressed. This information is indispensable to ensure that the experimental comparisons are grounded in validity.

- **Q6:** Several contemporary studies, including [R1], [R2], and [R3], harness SAM for segmenting objects from LiDAR point clouds. What distinct advantages does ACL offer when juxtaposed against these existing methodologies?

- **Q7:** Camera-LiDAR calibrations might not always achieve perfection. In scenarios where calibration discrepancies exist, there's potential for the SAR-derived coarse instance masks to be tainted with errors. The authors are suggested to consider conducting auxiliary experiments to probe whether calibration inaccuracies compromise the efficacy of the proposed method.

- **Q8:** This paper could be enriched with comparative analyses against some of the latest entrants in the domain of weakly and semi-supervised 3D object detectors.

- **Q9:** A minor formatting suggestion: For improved readability, perhaps anchor tables such as Table 2, Table 4, Table 5, Table7, and Table 10 to the top of their respective pages.

- **Q10:** This paper could benefit from having a standalone paragraph discussing the limitations and potential negative social impact of this work.

---

### Justification of the Rating

- The paper is overall well written with good insights. I am lean to upgrade the rating if the authors could resolve the concerns raised above.

---

### References
- [R1] D. Zhang, et al. “SAM3D: Zero-Shot 3D Object Detection via Segment Anything Model.” arXiv preprint arXiv:2306.02245.

- [R2] Y. Liu, et al. “Segment Any Point Cloud Sequences by Distilling Vision Foundation Models.” arXiv preprint arXiv:2306.09347.

- [R3] R. Chen, et al. “Towards Label-free Scene Understanding by Vision Foundation Models.” arXiv preprint arXiv:2306.03899.

**Details Of Ethics Concerns:**

No ethics concern observed.

---

> ### Author Response · Authors · 2023-11-14
> **Response to Reviewer joqH (Part 1)**
>
> We sincerely appreciate your questions and suggestions for our work. It requires considerable effort to meticulously read the paper and provide insightful and detailed questions and feedback. We believe these suggestions hold significant value for our work.
>
> > W1: A number of assertions lack rigorous justification or analytical backing.
>
> Thanks for your insightful suggestion regarding the need for rigorous justification and analytical backing. In response, we will enhance the manuscript to ensure a more comprehensive and transparent presentation of the underlying rationale. The revision will be reflected in both the responses to Questions and the upcoming edited version of the manuscript.
>
> > W2: The paper omits certain pivotal implementation specifics.
>
> Due to the page limit, we couldn't elaborate on all implementation details. However, we appreciate your suggestion, and we will carefully include detailed implementation of crucial steps. In addition, we commit to open-sourcing the code for reference in the future.
>
> > Q1: Elucidation about the term “data size” in pilot study.
>
> We sincerely apologize for the confusion caused by the term “data size”. We will replace it with “data amount” to convey that it refers to the amount of cropped regions used in training the SparseConv Detector, rather than the size of objects or instances. By changing the amount of training data, we demonstrate that geometry estimation does not necessitate a large amount of data.
>
> Moreover, we supplement experiments to enhance the claim that “massive data is necessary for semantic learning”. More details can be found in [Response to Reviewer yY5M (Part 1) W2](https://openreview.net/forum?id=Q1vkAhdI6j&noteId=rAHhKGSTQF).
>
> > Q2: The mechanics of the “clicking annotation”.
>
> We will address your questions regarding the mechanics of the “clicking annotation” as follows.
>
> - “Clicking annotation” for label-efficient 3D detection is first proposed by Meng et al. [R4], [R5]. They require annotators to click object centers in the BEV perspective, which are used to supervise the center prediction.
> - In the process of labeling, there is no difference made for different instance types.
> - For the last question about “Was manual labeling extended to all point clouds and how were the crucial clicking points discerned”. We are not sure we understand the reviewer’s question accurately. Our understanding is that the reviewer wants to know if the “click annotation” annotates all the objects in the training dataset. If we understand accurately, the answer is no. Using their implementation as an example, they randomly choose 500 frames from KITTI, where the centers of all instances are labeled in each frame. If our understanding is not accurate, we are open to any further discussions at anytime.
>
> Below we attach the comparison with Meng et al. [R4], [R5] to showcase our superiority. The evaluation results on KITTI val set (Car) with 534 box-level labels along with weakly labeled annotations.
>
> | Label-efficient method | Annotation | Easy | Moderate | Hard |
> |:---:|:---:|:---:|:---:|:---:|
> | WS3D [R4] | 534 boxes + weak labels | 84.04 | 75.10 | 73.29 |
> | WS3D [R5] | 534 boxes + weak labels | 85.04 | 75.94 | 74.38 |
> | MixSup | 534 boxes + weak labels | 86.37 | 76.20 | 72.36 |
>
> > Q3 & Q4: SAR and CCL
>
> Thank you for raising questions about contents that were not clearly elucidated. We sincerely apologize for any inconvenience caused by this. Since SAM's over-segmentation and nuScenes' imprecise point-pixel projection result in mediocre 3D instance masks, we leverage the spatial separability of objects in point clouds to refine the segmentation with Separability-Aware Refinement (SAR), which is demonstrated in the response to Q7. In SAR, the Connected Components Labeling (CCL) is a conventional operation using Euclidean distance for connectivity as outlined in https://docs.scipy.org/doc/scipy/reference/generated/scipy.sparse.csgraph.connected_components.html. Our code will be open-sourced for reference.
>
> > Q5: pre-trained semantic segmentation model
>
> The model we employed is the HTC [R6]  pre-trained on nuImages dataset. In automated coarse labeling, we solely utilized its semantic segmentation head to obtain semantics for each pixel. There is no need for concern regarding data leakage, as the author of nuScenes have explicitly stated that the overlap between nuImages and nuScenes is negligible in  https://github.com/nutonomy/nuscenes-devkit/issues/550#issuecomment-769571712

---

> ### Author Response · Authors · 2023-11-14
> **Response to Reviewer joqH (Part 2)**
>
> > Q6: Distinct advantages when juxtaposed against other methodologies utilizing SAM for segmenting objects from LiDAR point clouds.
>
> To the best of our knowledge, it is the first initiative harnessing SAM for instance segmentation in outdoor scenes. Notably, ACL achieves performance on par with the recent fully supervised panoptic segmentation models without the need for any 3D annotations. Compared to these methods, our Automated Coarse Labeling (ACL) innovatively leverages the inherent spatial separability of point clouds to refine instance segmentation, enabling the mitigation of the projection error (shown in response to Q7).
>
> In contrast, [R1] is confined to detecting the "car" category, while [R2] and [R3] focus on pretraining via pixel-point projection for semantic segmentation without addressing instance segmentation. It is crucial to emphasize that ACL is proposed to provide coarse labels for MixSup.
>
> We are delighted to see exceptional works like [R2] and [R3] playing a similar role and potentially integrating with MixSup. Finally, we appreciate your reminder and will add a detailed comparison with them in the revised version.
>
> > Q7: Inaccurate calibration
>
> We sincerely appreciate your suggestion for experiments regarding the inaccurate calibration. Throughout our experimentation, we identified inherent inaccuracies in the calibration of nuScenes, leading to discrepancies in the pixel-point projection of foreground objects. Hereby, we introduce SAR module to mitigate the degradation in segmentation caused by projection error.
>
> To further investigate the impact of projection discrepancies on automated coarse labeling, we introduced random noise to each camera's position. The performance for foreground objects on nuScenes val split panoptic segmentation is summarized in the table below. The results indicate that SAR-derived coarse instance masks exhibit a certain degree of resilience to calibration inaccuracies, attributed to the refinement introduced by the SAR module.
>
> | Noise of Camera Position (cm) | SAR | PQ | SQ | RQ |
> |---|:---:|:---:|:---:|:---:|
> | 0 | x | 44.9 | 74.6 | 59.8 |
> | -5~5 | x | 43.8 | 74.6 | 58.3 |
> | -10~10 | x | 41.4 | 74.5 | 55.2 |
> | 0 | √ | 63.7 | 82.6 | 76.9 |
> | -5~5 | √ | 62.8 | 82.3 | 76.1 |
> | -10~10 | √ | 60.6 | 81.4 | 74.3 |
>
> > Q8: Analyze some of the latest entrants in the domain of weakly and semi-supervised 3D object detectors
>
> - We highly value your suggestion and have enriched the comparisons with relevant works. To incorporate more entrants, we find a new work CoIn [R7] published in ICCV2023, with a 5% annotation cost on the KITTI validation split, evaluating Car-3D AP R40. It's worth noting that the 3.3% annotation cost for MixSup in the experiment is borrowed from other experiments. Despite having a lower annotation cost than CoIn, we achieve higher AP in both moderate and hard difficulties. We will include the comparison in the revised manuscript.
>
> | Annotation cost | Method | Detector | Easy | Moderate | Hard |
> |:---:|:---:|:---:|:---:|:---:|:---:|
> | 5% | CoIn[R7] | CenterPoint | 81.64 | 67.48 | 58.32 |
> | 3.3% | MixSup | CenterPoint | 80.02 | 69.12 | 65.62 |
>
> - It's also worthwhile to note that MixSup is orthogonal and complementary to semi-supervised learning, such as a simple self-training as highlighted in Sec 5.5 Table 6 in the paper. We will delve into the combination of MixSup and other semi-supervised training schemes in future work.
>
> > Q9: Formatting tables.
>
> Thank you for your generous suggestion. We will modify tables in the subsequent manuscript.
>
> > Q10 Discussing the limitations and potential negative social impact of this work.
>
> Regarding the limitations of MixSup, as a novel label-efficient paradigm, it is orthogonal to other label-efficient methods. Our work has not explored integration with semi-supervised methods, providing an avenue for potential performance enhancements. Moreover, our proposed automated coarse labeling, when combined with other exceptional 3D segmentation models, could yield higher-quality coarse labels. We believe MixSup holds significant promise in reducing annotation costs for the community, contributing to the conservation of both human and environmental resources. However, due to the cost savings in annotation come with a certain performance trade-off, practical deployment may raise risks of compromising driving safety.
>
> We are more than willing to continue engaging in in-depth discussions. If you have any further questions, please feel free to comment at any time.

---

> ### Author Response · Authors · 2023-11-14
> **Response to Reviewer joqH (Part 3)**
>
> ### **References**
>
> - [R1] Zhang, Dingyuan, et al. "SAM3D: Zero-Shot 3D Object Detection via Segment Anything Model." *arXiv preprint arXiv:2306.02245* (2023).
> - [R2] Liu, Youquan, et al. "Segment Any Point Cloud Sequences by Distilling Vision Foundation Models." *arXiv preprint arXiv:2306.09347* (2023).
> - [R3] Chen, Runnan, et al. "Towards label-free scene understanding by vision foundation models." *Thirty-seventh Conference on Neural Information Processing Systems*. 2023.
> - [R4] Meng, Qinghao, et al. "Weakly supervised 3d object detection from lidar point cloud." *European Conference on computer vision*. Cham: Springer International Publishing, 2020.
> - [R5] Meng, Qinghao, et al. "Towards a weakly supervised framework for 3d point cloud object detection and annotation." *IEEE Transactions on Pattern Analysis and Machine Intelligence* 44.8 (2021): 4454-4468.
> - [R6] Chen, Kai, et al. "Hybrid task cascade for instance segmentation." *Proceedings of the IEEE/CVF conference on computer vision and pattern recognition*. 2019.
> - [R7] Xia, Qiming, et al. "CoIn: Contrastive Instance Feature Mining for Outdoor 3D Object Detection with Very Limited Annotations." *Proceedings of the IEEE/CVF International Conference on Computer Vision*. 2023.

---

> ### Author Response · Authors · 2023-11-18
> **Update to the response to Q2 and Q8**
>
> We have updated the response to Q2 and Q8. We are open to any further discussions at any time.

---

> > ### Comment · Reviewer_joqH · 2023-11-18
> >
> > I would like to thank the authors for their efforts during the rebuttal.
> >
> > I have read the authors' responses, as well as other reviewers' comments and authors' responses to them, It is glad to see that many of the unclear statements have been addressed, some missing information have been supplemented, and additional experimental results have been provided during the rebuttal.
> >
> > I have an overall favorable opinion and will upgrade the rating. The authors are suggested to incorporate revisions and new updates in their revised manuscript.

---

### Official Review · Reviewer_yY5M · 2023-10-29

**Soundness:** 3 good
**Presentation:** 2 fair
**Contribution:** 2 fair
**Rating:** 6
**Confidence:** 4

**Summary:**

The authors proposed MixSup for efficient LiDAR-based 3D object detection. It mainly consists of two contributions: (1) a cluster-level coarse labeling scheme by just labeling the 3 bounding box corners in bird's eye view, which the authors claim to have only 14% labeling time compared to full 3D labeling; (2) a learning strategy that utilizes both few full 3D labels and many coarse labels by training classification / proposal stage with only coarse labels and regression stage with full 3D labels. The authors show that on mulitple dataset and with multiple different detection mdoels, the proposed method can achieve the performance of the full supervised counterpart using all full 3D labels.

**Strengths:**

- The paper is overall easy to follow.
- The paper focuses on efficient learning for 3D detection, especially with an emphasis on autonomous driving applications. This active subarea holds significant promise to reduce the cost in autonomous vehicles.
- The proposed method is overall simple but with good performance in various settings.
- The authors also explored the automatic coarse labeling setting utilizing a SAM model.

**Weaknesses:**

1. Discussing and comparing with [Meng et al. 2020; 2021]. As pointed out in the related works in the paper, there are a bunch of existing works trying to reduce the labeling cost in the 3D detection task, and one of the closest works to this paper is probably [Meng et al. 2020; 2021], where they explored a similar setup: training models with many weakly labeled and few fully labeled 3D data. As a reader, I would expect to see further discussion and comparison between the proposed work and this work (e.g. the difference in labeling cost, the performance difference, and why the proposed method is a better approach). I do read that in Sec 4.5 and the main results, the authors did compare the proposed method with some of the label-efficient frameworks, but it looks like the comparison with [Meng et al. 2020; 2021] is missed out.

2. The hypothesis is not fully backed up: in the last paragraph of Sec 3 (pilot study), the authors concluded: "This phenomenon suggests that LiDAR-based detectors indeed only need a very limited number of accurate labels for geometry estimation. Massive data is only necessary for semantic learning." If I understand correctly, the pilot experiments only provide evidence to some degree in the first half, but the latter half "Massive data is only necessary for semantic learning" seems not fully grounded. An evaluation of how well the model can perform with different amounts of semantic labels could better support the claim.

3. The study on the labeling costs is not fully grounded. In Sec 5.3, the authors claim "The average time cost of a coarse cluster label is only 14% of an accurate box label." In Sec 4.2, the authors also claim "In addition, it is also non-trivial for annotators to make an accurate center click." However, evidence is missing for these claims. Where does the "14% cost" come from? How much harder / more inaccurate is it for the annotator to click the center? Do you perform a user study with a reasonable number of annotators? How similar are the coarse labels from the annotators to the simulated coarse labels used in the experiments? I would suggest the authors include more details to back the claim.

4. Presentation:

    a) I would suggest the authors be more specific about "semantic" and "geometry" from the beginning of the paper. If I understand correctly, by "semantic" the authors meant to coarsely identify object locations and types, and by "geometry" they meant to accurately regress the discovered object's location, dimensions, and heading. It is a bit confusing when reading the 3rd paragraph of the introduction.

    b) The writing in Sec 4 is somewhat confusing and unclear. Firstly, Figure 3 is not referred from anywhere in the main text and the caption is not self-contained, resulting in little help for understanding the idea. And Sec 4.2 is confusing: what does it mean by "assignment"? If I understand correctly after carefully reading into the latter parts, what the authors meant was that to properly supervise the proposal/classification stage for the detector. It is not self-contained in this sense, i.e. a reader without good knowledge of how these detectors are designed will have an even harder time understanding the method.


[Meng et al. 2020] Qinghao Meng, Wenguan Wang, Tianfei Zhou, Jianbing Shen, Luc Van Gool, and Dengxin Dai.
Weakly supervised 3d object detection from lidar point cloud. In ECCV, pp. 515–531. Springer, 2020.
[Meng et al. 2021] Qinghao Meng, Wenguan Wang, Tianfei Zhou, Jianbing Shen, Yunde Jia, and Luc Van Gool. Towards a weakly supervised framework for 3d point cloud object detection and annotation. IEEE transactions on pattern analysis and machine intelligence, 44(8):4454–4468, 2021.
[Liang et al. 2021] Liang, H., Jiang, C., Feng, D., Chen, X., Xu, H., Liang, X., ... & Van Gool, L. (2021). Exploring geometry-aware contrast and clustering harmonization for self-supervised 3d object detection. In Proceedings of the IEEE/CVF International Conference on Computer Vision (pp. 3293-3302).

**Questions:**

Please see the weaknesses section. Additionally,

1. In Table 4, it looks the PointContrast and the ProposalContrast have negative performance gains compared with training with 10% frames. But from the GCC-3D paper [Liang et al. 2021], the PointContrast usually can improve over the from-scratch baseline. The results reported here look inconsistent with previous literature. I am wondering how are these baselines trained?

---

> ### Author Response · Authors · 2023-11-17
> **Response to Reviewer yY5M (Part 1)**
>
> We sincerely appreciate your insightful questions and we believe they hold significant value for our work. We try to resolve your concerns below.
>
> > W1: Further discussion and comparison between the proposed work [R1, R2] and this work.
>
> We apologize for unintentionally overlooking the comparison as our focus was primarily on Waymo and nuScenes. Here we supplement a discussion and experimental comparison with **WS3D** [Meng et al. 2020/2021] as follows.
>
> - WS3D has a specifically designed detection pipeline and cannot be generalized to other detectors. In contrast, MixSup can be integrated with various detectors.
> - The coarse cluster-level labels proposed in our method can be obtained through foundational models such as SAM, as we demonstrated in Automated Coarse Labeling.
> - For the annotation cost, we are truly sorry that we cannot provide a rigorous quantitative comparison with center-click in WS3D because the cost largely depends on labeling tools and protocols, but we do not have the labeling tools in WS3D. However, we provide the analysis and discussion in the following **W3**.
> - At the same annotation cost, our method has better performance. In the following Table, we report the performance of Car detection in KITTI val split since WS3D does not use Waymo and nuScenes dataset.
>
> | Label-efficient method | Annotation | Easy | Moderate | Hard |
> |:---:|:---:|:---:|:---:|:---:|
> | WS3D [R1] | 534 boxes + weak labels | 84.04 | 75.10 | 73.29 |
> | WS3D [R2] | 534 boxes + weak labels | 85.04 | 75.94 | 74.38 |
> | MixSup | 534 boxes + weak labels | 86.37 | 76.20 | 72.36 |
>
> > W2: "Massive data is only necessary for semantic learning" seems not fully grounded.
>
> We greatly appreciate your insightful feedback. We supplement the experiments to enhance this claim.
>
> Specifically, we decrease the data amount (#. training frames) from 100% to 10% in Waymo and train a popular detector CenterPoint [R7]. Such a decrease in data amount has a negative impact on **both** geometry learning and semantic learning. To mitigate the impact on geometry learning and only focus on semantic learning, we aggressively relax the IoU thresholds in evaluation from 0.7 / 0.5 / 0.5 to 0.5 / 0.25 / 0.25 for vehicle/pedestrian/cyclist, respectively.
>
> Let’s first look at the following table, there are huge performance gaps between using 100% data and using 10% data. Obviously, here these gaps are caused by the degradation of **both** semantic learning and geometry estimation.
>
> | Data amount | IoU thresholds | Vehicle L2 AP / APH | Pedestrian L2 AP / APH | Cyclist L2 AP / APH |
> |---|:---:|:---:|:---:|:---:|
> | 100% | 0.7 / 0.5 / 0.5 (Normal) | 65.42 / 64.92 | 66.49 / 60.53 | 69.28 / 68.12 |
> | 10% | 0.7 / 0.5 / 0.5 (Normal) | 55.92 / 55.39 | 56.97 / 49.07 | 51.52 / 50.46 |
> | Performance Gaps | - | 9.50 / 9.53 | 9.52 / 11.46 | 17.76 / 17.66 |
>
> Then let’s look at the following table, where we **relax the IoU thresholds for evaluation** to mitigate the negative impact introduced by degradation of geometry estimation. The overall performance becomes much higher after relaxing IoU thresholds. However, there are still significant gaps between using 100% data and using 10% data. Especially, for Pedestrian and Cyclist, the gaps (between 10%
>  and 100%) **do not even get smaller** after we relax the IoU thresholds. Thus, the performance gaps between 100% data and 10% data **should be mainly caused by the degradation of semantic learning.** In other words, semantic learning is sensitive to the data amount.
>
> | Data amount | IoU thresholds | Vehicle L2 AP / APH | Pedestrian L2 AP / APH | Cyclist L2 AP / APH |
> |---|:---:|:---:|:---:|:---:|
> | 100% | 0.5 / 0.25 / 0.25 (relaxed) | 87.15 / 86.20 | 82.99 / 74.62 | 74.01 / 72.70 |
> | 10% | 0.5 / 0.25 / 0.25 (relaxed) | 81.99 / 80.67 | 73.61 / 62.14 | 56.50 / 55.20 |
> | Performance Gaps | - | 5.16 / 5.53 | 9.38 / 12.48 | 17.51 / 17.50 |
>
> From another point of view, in the main paper, we have decreased the data amount of well-classified patches from 100% to 10% to reveal the impact on geometry estimation, as shown in the following table. Compared with the aforementioned performance change caused by semantic learning degradation, **the performance change in the following table is negligible**. Thus, we draw the conclusion that “LiDAR-based detectors indeed only need a very limited number of accurate labels for geometry estimation. Massive data is only necessary for semantic learning”.
>
> *The following table: Performances with varying data amounts on the well-classified dataset (Copy from main paper).*
>
> | Data amount | IoU thresholds for three classes | Vehicle L2 AP / APH | Pedestrian L2 AP / APH | Cyclist L2 AP / APH |
> |---|:---:|:---:|:---:|:---:|
> | 100% | 0.7 / 0.5 / 0.5 | 64.19 / 63.74 | 65.23 / 58.02 | 67.04 / 65.99 |
> | 10% | 0.7 / 0.5 / 0.5 | 63.37 / 62.89 | 64.78 / 57.96 | 66.26 / 65.14 |
> | Performance Gaps | - | 0.82 / 0.85 | 0.45 / 0.06 | 0.78 / 0.85 |
>
> The discussion above will be included in the edited version.

---

> ### Author Response · Authors · 2023-11-17
> **Response to Reviewer yY5M (Part 2)**
>
> > W3: The study on the labeling costs is not fully grounded.
>
> We highly value your questions and try to resolve your reasonable concerns from the following aspects.
>
> - As described in Sec 5.3, we conducted a study by asking **three** experienced annotators to label 100 frames from different sequences of nuScenes. “The average time cost of a coarse cluster label is only 14% of an accurate box label” is based on the statistics gathered from the annotators.
> - In the original paper, we simulate the coarse labels by randomly expanding GT boxes 0~10% in each dimension without evaluating the similarity with human coarse annotation. Thank you so much for pointing this out! To be more rigorous, here we first evaluate the similarity between human-annotated coarse labels (100-frame subset) and ground-truth labels, **which is 81.22% in terms of segmentation mIoU**. Then we add noise to ground-truth labels to simulate more rigorous coarse labels, controlling coarse labels possessing **80% to 82%** mIoU with ground-truth labels, just like the human-annotated coarse labels. In particular, we use the following two noises:
>     - Randomly shifting center locations $-0.1\sim 0.1$ m and expanding 0%$\sim$50% in each dimension (Noise ①). This noise has 81.91% mIoU with the ground-truth label.
>     - Randomly shifting center locations $-0.2\sim 0.2$ m, expanding 0%$\sim$ 20%in each dimension, and rotating $-15^\circ\sim15^\circ$ in heading (Noise ②). This noise has 80.60% mIoU with the ground-truth label.
>
>     We then conduct experiments on nuScenes using such noisy coarse labels, results shown in the following table. Performance with noise ①, ② is similar to the performance in the original paper. Thus, our simulated coarse labels are reliable for the experiments in our paper.
>
> | Detector | Noise | mAP | NDS |
> |---|---|---|---|
> | CenterPoint (100%) | - | 62.41 | 68.20 |
> | CenterPoint (MixSup) | original paper | 60.73 | 66.46 |
> | CenterPoint (MixSup) | Noise ① | 60.23 | 65.99 |
> | CenterPoint (MixSup) | Noise ② | 60.21 | 66.28 |
>
> - As for the center-click annotation, we unintentionally overlooked the human labeling study because it is not a core part of our method. We cannot launch and finish a labeling study during the limited rebuttal period without the labeling tools in WS3D. We are truly sorry for that. However, we demonstrate that center-click annotation has some limitations.
>     - For the inevitable instances with very partial point clouds, accurately clicking the center is challenging.  Conversely, MixSup's cluster-level label only requires clicking around the visible part of an object with three points, ensuring that the generated parallelogram encompasses the partial point clouds.
>     - Some methods need more than centers, such as the recent state-of-the-art FSD [R3]. It adopts a segmentation network in the first stage to obtain foreground points, **requiring point-level supervision**. Here we attach the performance on Waymo validation split with L2 APH using FSD as the base detector to show the effectiveness of MixSup, where the percentage in parentheses indicates the performance ratio to the fully supervised counterpart.
>
> | Detector | Mean | Vehicle | Pedestrian | Cyclist |
> |---|---|---|---|---|
> | FSD (100% frames) | 71.27 | 70.09 | 69.79 | 73.93 |
> | FSD (MixSup) | 68.57 (96.21%) | 66.08 (94.28%) | 66.53 (95.33%) | 73.09 (98.86%) |
>
> > W4: Presentation.
>
> We sincerely appreciate your meticulous reading of our work. Your understanding of "semantic," "geometry," and "assignment" is highly accurate. We definitely believe good papers necessitate high-quality writing and are truly sorry for the unclear expressions. We highly value your suggestions and will make revisions in the upcoming manuscript.
>
> > Q1: PointContrast and the ProposalContrast have negative performance gains.
>
> We apologize for any confusion caused. In our paper, we took a recent method MV-JAR [R4] (CVPR 23) as a compared method. The results of PointContrast and ProposalContrast in our paper are reported by MV-JAR.  For a direct comparison, we compare MixSup with the results in the **original ProposalConstrast paper** as follows. Using **1% boxes and coarse labels**, MixSup is slightly better than them with **20% frame annotations**. Due to the different settings, we cannot perfectly align the annotation cost with them, but the following results suggest MixSup could achieve promising results.
> | Detector | annotation cost |Mean L2 APH (Waymo val) |
> |---|---|---|
> | PVRCNN + GCC 3D [R6] | 20% frames |58.18 |
> | PVRCNN + ProposalContrast |  20% frames | 59.28 |
> | PVRCNN + MixSup (ours) | 1% boxes + coarse labels | 59.74 |
>
> We are more than willing to continue engaging in in-depth discussions. If you have any further questions, please feel free to comment at any time.

---

> ### Author Response · Authors · 2023-11-17
> **Response to Reviewer yY5M (Part 3)**
>
> ### **References**
>
> - [R1] Meng, Qinghao, et al. "Weakly supervised 3d object detection from lidar point cloud." *European Conference on computer vision*. Cham: Springer International Publishing, 2020.
> - [R2] Meng, Qinghao, et al. "Towards a weakly supervised framework for 3d point cloud object detection and annotation." *IEEE Transactions on Pattern Analysis and Machine Intelligence* 44.8 (2021): 4454-4468.
> - [R3] Fan, Lue, et al. "Fully sparse 3d object detection." *Advances in Neural Information Processing Systems* 35 (2022): 351-363.
> - [R4] Xu, Runsen, et al. "MV-JAR: Masked Voxel Jigsaw and Reconstruction for LiDAR-Based Self-Supervised Pre-Training." *Proceedings of the IEEE/CVF Conference on Computer Vision and Pattern Recognition*. 2023.
> - [R5] Fan, Lue, et al. "Embracing single stride 3d object detector with sparse transformer." *Proceedings of the IEEE/CVF conference on computer vision and pattern recognition*. 2022.
> - [R6] Liang, Hanxue, et al. "Exploring geometry-aware contrast and clustering harmonization for self-supervised 3d object detection." *Proceedings of the IEEE/CVF International Conference on Computer Vision*. 2021.
> - [R7] Yin, Tianwei, et al. "Center-based 3d object detection and tracking." *Proceedings of the IEEE/CVF conference on computer vision and pattern recognition*. 2021.

---

> > ### Comment · Reviewer_yY5M · 2023-11-21
> >
> > Thank the authors for carrying out the rebuttal!
> >
> > I have read through the rebuttal and comments from other reviewers. The response resolved most of my concerns thus I raised the rating to leaning to acceptance. The authors are suggested to include the discussion and analysis in the final version.

---

### Author Response · Authors · 2023-11-21
**Paper revision uploaded and looking forward to further discussion**

Dear reviewers and ACs,

We sincerely appreciate your dedicated efforts in the review process!  The constructive suggestions definitely make our manuscript better! We are pleased that the reviewers highlight our paper as having “**significant promise to reduce the cost**”,  being “**firmly rooted in a well-articulated motivation and logical progression**”, “**with good performance in various settings**”, “**intuitive and efficient**”, “**well written and pleasant to read**”, and “**illustrations are well-done**”.

We have uploaded the revised manuscript and added most of the revisions and discussions in the main text and the appendix.

Considering the official reviewing guidance, which states that “unlike previous years, there will be **no second stage** of discussion between authors and reviewers” and the current discussion stage will last until **Nov 22**, we are looking forward to participating in further discussion with the reviewers to enhance the quality of our work.

Best Regards

Authors

---

### Meta-Review · Area_Chair_zqma · 2023-12-06

**Metareview:**

All the three reviewers upgraded the rating to positive after rebuttal. In the original reviews, there are main concerns about 1) comparisons with existing work; 2) more discussions about the labeling cost; 3) some technical clarity. In the rebuttal, the authors have addressed most issues and appreciate the contributions. Hence, the acceptance rating is recommended, in which the authors should incorporate the comments from reviewers in the final version.

**Justification For Why Not Higher Score:**

Although all the reviews are positive after rebuttal, there are still many critical analysis and discussions that are needed to include in the paper.

**Justification For Why Not Lower Score:**

N/A

---

### Decision · Program_Chairs · 2024-01-16

Accept (poster)